# Foveal vision anticipates defining features of eye movement targets

**Lisa M Kroell**[1,2]*, **Martin Rolfs**[1,2,3,4]

[1]Department of Psychology, Humboldt-Universität zu Berlin, Berlin, Germany; [2]Berlin School of Mind and Brain, Humboldt-Universität zu Berlin, Berlin, Germany; [3]Exzellenzcluster Science of Intelligence, Technische Universität Berlin, Berlin, Germany; [4]Bernstein Center for Computational Neuroscience Berlin, Berlin, Germany

**Abstract** High-acuity foveal processing is vital for human vision. Nonetheless, little is known about how the preparation of large-scale rapid eye movements (saccades) affects visual sensitivity in the center of gaze. Based on findings from passive fixation tasks, we hypothesized that during saccade preparation, foveal processing anticipates soon-to-be fixated visual features. Using a dynamic large-field noise paradigm, we indeed demonstrate that defining features of an eye movement target are enhanced in the pre-saccadic center of gaze. Enhancement manifested as higher Hit Rates for foveal probes with target-congruent orientation and a sensitization to incidental, target-like orientation information in foveally presented noise. Enhancement was spatially confined to the center of gaze and its immediate vicinity, even after parafoveal task performance had been raised to a foveal level. Moreover, foveal enhancement during saccade preparation was more pronounced and developed faster than enhancement during passive fixation. Based on these findings, we suggest a crucial contribution of foveal processing to trans-saccadic visual continuity: Foveal processing of saccade targets commences before the movement is executed and thereby enables a seamless transition once the center of gaze reaches the target.

*For correspondence: lisa.maria.kroell@hu-berlin.de

**Competing interest:** The authors declare that no competing interests exist.

## Editor's evaluation

In a methodologically sophisticated study of pre-saccadic processing at the fovea, Kroell and Rolfs provide compelling evidence that saccade preparation causes feature-specific pre-saccadic visual enhancement restricted largely to the center of gaze. The authors were able to differentiate this effect from pre-saccadic enhancement during passive fixations and to rule out criterion shifts as a mechanistic explanation. The fundamental implication of these findings will be of interest to both vision scientists and modelers. They parametrize a potential mechanism for visual continuity across saccades, with foveal processing identified as a key, contributing component.

## Introduction

Foveal processing is of singular importance for primate vision. Within a small, central region of the retina known as the foveal pit, retinal layers spread aside and allow light to impinge directly onto a densely packed population of cone photoreceptors – the foveola (*Hageman and Johnson, 1991*). The resulting signals carry highly resolved visual information, the prioritization of which is reflected in the organization of upstream neural areas: even though the fovea covers merely the central 5 degrees of the visual field, more than 40% of primary visual cortex is devoted to the processing of foveal input (*Curcio et al., 1990*; *Tootell et al., 1988*; *Hendrickson, 2005*). To utilize the resolution of those signals, primates routinely and rapidly move their eyes to bring relevant information into foveal vision.

Quite surprisingly, foveal processing appears understudied on both a neurophysiological and behavioral level (*Schira et al., 2009*). In the moving observer in particular, decades of research have characterized pre-saccadic sensitivity modulations at the target of eye movements (*Zhao et al., 2012*; *Squire et al., 2013*; *Li et al., 2021*). To date, little to nothing is known about the concurrent development of visual sensitivity in the pre-saccadic center of gaze (*Hanning and Deubel, 2022*; *Ludwig et al., 2014*). We hypothesized that the fovea is not simply a passive receiver of input for high-acuity vision. Instead, it appears uniquely suited to predict incoming information and, in consequence, to assume a crucial role in the establishment of visual continuity across saccades.

Evidence from neuroimaging (*Williams et al., 2008*; *Fan et al., 2016*), brain stimulation (*Chambers et al., 2013*), and psychophysics (*Fan et al., 2016*; *Yu and Shim, 2016*; *Weldon et al., 2020*) suggests that during fixation, the fovea contributes to the processing of stimuli presented in the periphery: using functional magnetic resonance imaging, *Williams et al., 2008*; *Fan et al., 2016* demonstrate that relevant peripheral objects are represented in foveal retinotopic cortex, possibly via feedback connections from temporal areas (*Rockland and Van Hoesen, 1994*; *Bullier, 2001*). Foveal cortex thus seems to be recruited for tasks requiring high perceptual scrutiny – even when the respective stimulus appears far outside the receptive field of any foveal neuron. Indeed, disrupting foveal processing through transcranial magnetic stimulation (*Chambers et al., 2013*) or the presentation of a foveal distractor (*Fan et al., 2016*; *Weldon et al., 2020*) impairs peripheral discrimination performance. Foveal feedback, so far characterized during fixation, gains a predictive nature when applied to active vision: already before an eye movement, critical features of the peripheral target should be available for foveal processing. Once gaze has shifted, the target is foveated. In the moving observer, foveal feedback therefore anticipates incoming information and may facilitate post-saccadic target processing or gaze correction when the eyes land erroneously off-target (*Deubel et al., 1982*; *Ohl and Kliegl, 2016*; *Tian et al., 2013*). Most notably, this mechanism would not require predictive spatial updating to support transsaccadic continuity: irrespective of the future foveal location, feedback to the current center of gaze would suffice to predict the features of the eye movement target in retinotopic coordinates.

Despite the theoretical usefulness of such a mechanism, there are reasons to assume that foveal feedback as characterized during fixation may break down while an eye movement is prepared to a different visual field location. First and foremost, saccade preparation is accompanied with an obligatory shift of attention to the saccade target (*Zhao et al., 2012*; *Squire et al., 2013*; *Li et al., 2021*) which in turn has been shown to decrease foveal sensitivity (*Hanning and Deubel, 2022*; *Ludwig et al., 2014*). Moreover, the execution of a rapid eye movement induces brief motion signals on the retina (*Castet et al., 2002*) which may mask or in other ways interfere with the pre-saccadic prediction signal. On a more conceptual level, the recruitment of foveal processing as an 'active blackboard' (*Williams et al., 2008*; *Roelfsema and de Lange, 2016*) may become obsolete in the face of an imminent foveation of relevant peripheral stimuli – unless, of course, foveal processing serves the establishment of trans-saccadic visual continuity.

To shed light on this question, we characterized the immediate perceptual signature of foveal feedback during eye movement preparation. We hypothesized that, if foveal feedback remains effective during saccade preparation, fed-back information on the saccade target should combine with foveal input and thereby facilitate the detection of target-congruent, foveal feature information. On a perceptual level, this should correspond to a predictive enhancement of saccade target features in the pre-saccadic center of gaze.

The lack of knowledge on foveal processing is mostly due to methodological constraints: in neurophysiological investigations, foveal receptive fields are challenging to estimate since even in paralyzed monkeys, gaze is never fully stable (*Forte et al., 2002*). Only recent developments combining a free viewing approach with an offline reconstruction of visual input allow for these estimations (*Yates et al., 2021*). Psychophysical investigations of foveal processing require a measure sensitive enough to reveal subtle variations of high-acuity vision in behavioral responses. At the same time, the stimulus probing performance in the center of gaze must be inconspicuous enough not to interfere with saccade programming (*Rolfs et al., 2011*; *Hanning et al., 2019*). In light of these considerations, we smoothly embedded our stimuli in a dynamic stream of full-screen, 1 /f noise images (*Hanning and Deubel, 2022*; *Hanning et al., 2019*; *Hanning and Deubel, 2021*; see Materials and methods). Observers maintained fixation in the screen center while the images flickered at a temporal frequency

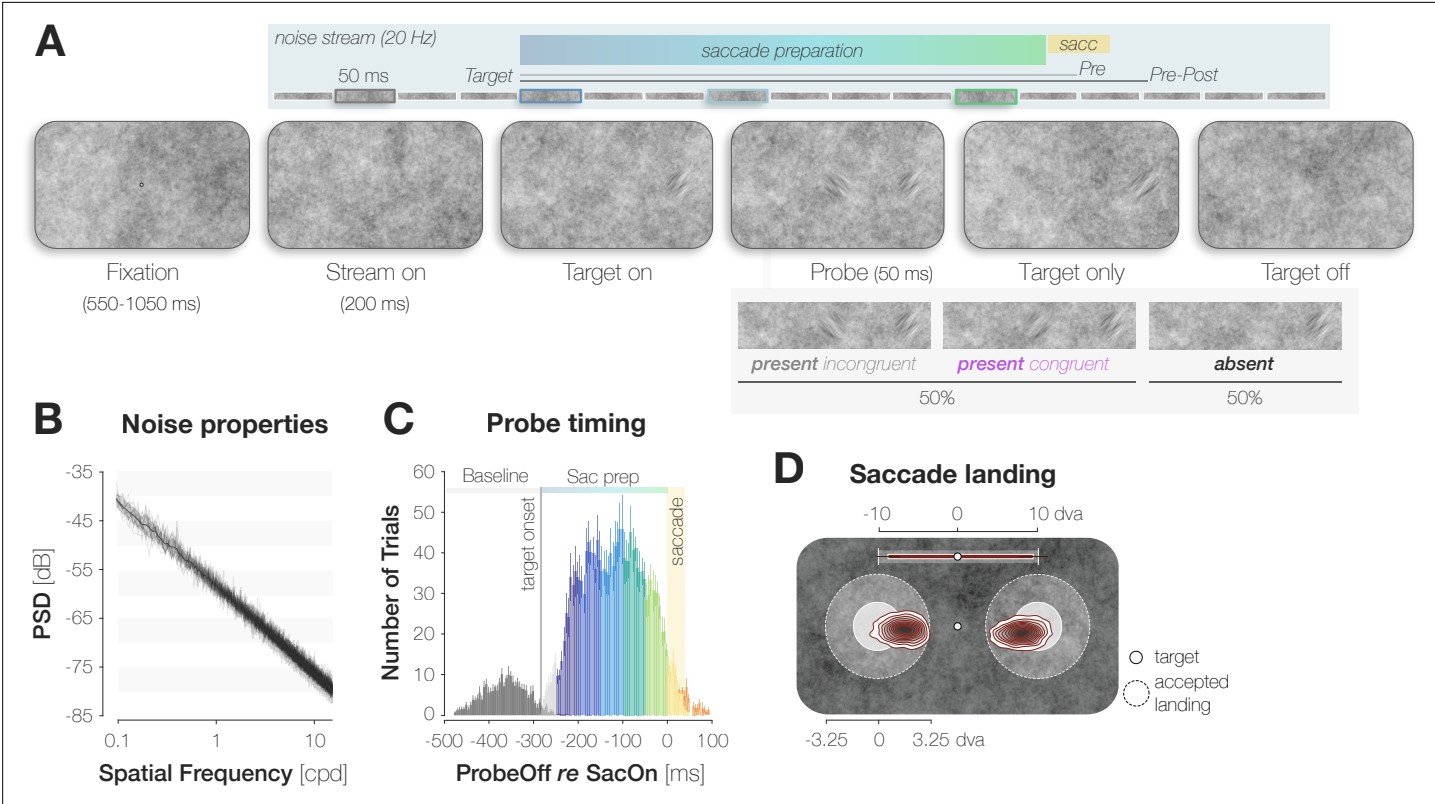

**Figure 1.** A dynamic noise paradigm probing foveal sensitivity to saccade target features. (**A**) Example trial procedure in Experiment 1: The saccade target and foveal probe were embedded in full-screen noise images flickering at a frequency of 20 Hz (image duration of 50ms). After 200ms, the saccade target (an orientation-filtered patch; filtered to either –45° or +45°; 3 dva in diameter) appeared 10 dva to the left or the right of the screen center, cueing the eye movement. On 50% of trials, a probe (a second orientation-filtered patch; filtered to either –45° or +45°) appeared in the screen center either 150ms before target onset (top panel; highlighted element with grey outline), or at an early (dark blue outline), medium (light blue outline) or late (green outline) stage of saccade preparation. The foveal probe was presented for 50ms and could be oriented either congruently or incongruently to the target. On *Pre* trials, the saccade target disappeared before saccade landing. On *Pre-Post* trials, it remained visible for a brief duration after landing. Observers reported if they had perceived the probe in the screen center or not (present vs absent). After a 'present' response, they reported the probe's perceived orientation (left for –45° vs right for +45°). (**B**) Noise properties: Power spectral density (PSD) of the foveal region (3 dva diameter) of all noise images presented in a randomly chosen experimental session. (**C**) Probe timing: histogram of time intervals between probe offset and saccade onset. Bar heights and error bars indicate the mean and standard error of the mean (SEM; n=7) across observers, respectively. On baseline trials, the probe appeared before target onset (dark grey bars). On all remaining trials, the probe appeared after target onset and therefore during saccade preparation (Sac prep). We assigned saccade preparation trials to five distinct time bins (from dark blue to light green). Trials in which the probe disappeared more than 250ms before saccade onset (light grey), during the saccade (yellow) or after saccade offset (orange) were excluded. The yellow background rectangle illustrates the median saccade duration. (**D**) Observers were able to select the peripheral stimulus as the saccade target: Bivariate Gaussian Kernel densities of saccade landing coordinates for left- and rightwards saccades. Filled circles indicate the saccade target (rad = 1.5 dva). Transparent circles indicate the accepted landing area (rad = 3.25 dva). The distance between the screen center and the targets was reduced for illustration purposes. Red bars on top indicate median saccade amplitudes based on both the horizontal and vertical component of the saccade. Neither saccade latencies nor saccade endpoints influenced congruency effects (see Materials and methods).

of 20 Hz (*Figure 1A*). At some point, the saccade target appeared 10 degrees of visual angle (dva) to the left or right of fixation, cueing the eye movement. The target was created by filtering the background noise at the respective location to an orientation of –45° or +45°. On 50% of trials, a probe stimulus, that is, an additional orientation-filtered noise patch, appeared in the screen center during saccade preparation and remained visible for 50ms. Crucially, the probe was oriented either –45° or +45° and therefore either congruently or incongruently to the target. After executing the saccade, observers reported if they had perceived a stimulus in the screen center (present versus absent). After responding 'present', they additionally reported the perceived orientation (left versus right for –45° and +45°, respectively). We investigated whether the orientation of the saccade target influenced the foveal detection judgment.

We made three main observations (*Figure 2B–D*). First, observers' responses suggest a foveal sensitization to the target orientation. Hit Rates (HRs) for target-congruent foveal probes started to exceed incongruent ones 175ms before saccade onset. In our design, the foveal region was never void of signal but contained incidental orientation information in the background noise even on probe absent trials. On those trials, we expected observers to become sensitive to target-congruent orientations in the foveal noise and, in consequence, to report the target orientation more often than the non-target orientation when generating a False Alarm (FA). Indeed, congruent False Alarm rates (FARs) exceeded incongruent ones. Second, just like the increase in HRs, the increase in FARs signifies *enhancement*. Target-congruent and incongruent FAs indeed relied on an incidental, high energy of the reported orientation in the foveal noise. Third, enhancement is *foveal* rather than global. Enhancement was most pronounced in the center of gaze and exhibited an asymmetric profile, extending further towards than away from the target. Based on these findings, we suggest that their predictive potential in active visual settings may constitute the key functionality of foveal feedback connections that, so far, have exclusively been characterized during passive fixation.

## Terminology

The foveola covers the central 1.3 degrees of visual angle (dva) (*Hendrickson, 2005*). The fovea and parafovea cover the central 5.5 and 8.3 dva, respectively (*Hendrickson, 2005*). Since our probe stimulus exhibited a diameter of 3 dva and therefore extended past the foveola and into the surrounding foveal region, we use the term 'fovea' to refer to observers' center of gaze throughout the article. To facilitate the integration of our findings into existing literature (*Stewart et al., 2020*), visual field locations outside the parafoveal area will be referred to as 'peripheral'.

## Results

### Hit and False Alarm rates suggest foveal enhancement of the target orientation

We defined congruent and incongruent HRs as the proportion of probe-present trials in which observers reported perceiving the probe, and the probe was oriented congruently and incongruently to the target, respectively (*Figure 2A*). Based on the time interval between the offset of the probe and the onset of the saccade, we assigned each Hit trial to one of five pre-saccadic time bins of 50ms duration (*Figure 1C*). We obtained a baseline sensitivity estimate outside the saccade preparation period by presenting the probe 150ms *before* the saccade target on an additional subset of trials.

A two-way repeated measures ANOVA on HRs revealed significant main effects of probe offset ($F(5,30) = 8.1$, p<0.001) and congruency ($F(1,6) = 9.3$, p=0.023), as well as an interaction, $F(5,30) = 3.3$, p=0.017. Irrespective of congruency, foveal HRs decreased continuously in the course of saccade preparation, when attention is known to shift to the target (*Hanning and Deubel, 2022*; *Ludwig et al., 2014*; *Figure 2B*; see *Appendix 1—figure 1* for individual observer data). We observed a HR of 86.5 ± 11.1% in the earliest pre-saccadic bin 250–200ms before saccade onset and a significantly lower HR of 77.1 ± 6.8% 50–0ms before the saccade (bootstrapped p<0.001; see Materials and methods). Congruent and incongruent HRs did not differ in the baseline ($HR_{cong-incong} = -0.7 ± 8.5\%$; p=0.589) and earliest pre-saccadic bin ($HR_{cong-incong} = -0.7 ± 5.1\%$; p=0.669). Afterwards, a stable congruency effect emerged: HRs for congruent probes significantly exceeded HRs for incongruent probes throughout the remaining bins ($HR_{cong-incong}$=6.0 ± 5.7%, 7.0 ± 6.5%, 6.4 ± 3.8%, 5.9 ± 5.2%, for the time bins [–200 –150[, [–150 –100[, [–100 –50[, [–50 0[ ms, respectively; all ps <0.003).

We defined congruent and incongruent FARs as the proportion of probe-absent trials in which observers reported perceiving the probe and generated a target-congruent and target-incongruent orientation report, respectively (*Figure 2A*). When generating a FA, observers reported the target orientation more often than the non-target orientation, leading to higher congruent than incongruent FARs ($FAR_{cong-incong}$=5.9 ± 3.3%; p<0.001; *Figure 2B*). FAs are not time-resolved since observers may have perceived the probe at any time in the course of a probe-absent trial.

Note that we defined Hits as 'present' responses on probe present trials. We did not take the accuracy of the subsequent orientation report into account since the difficulty of the foveal detection task had been adjusted to yield optimal performance levels for the presence/absence – not the orientation – judgment (see Materials and methods). Nonetheless, only including 'present' responses after

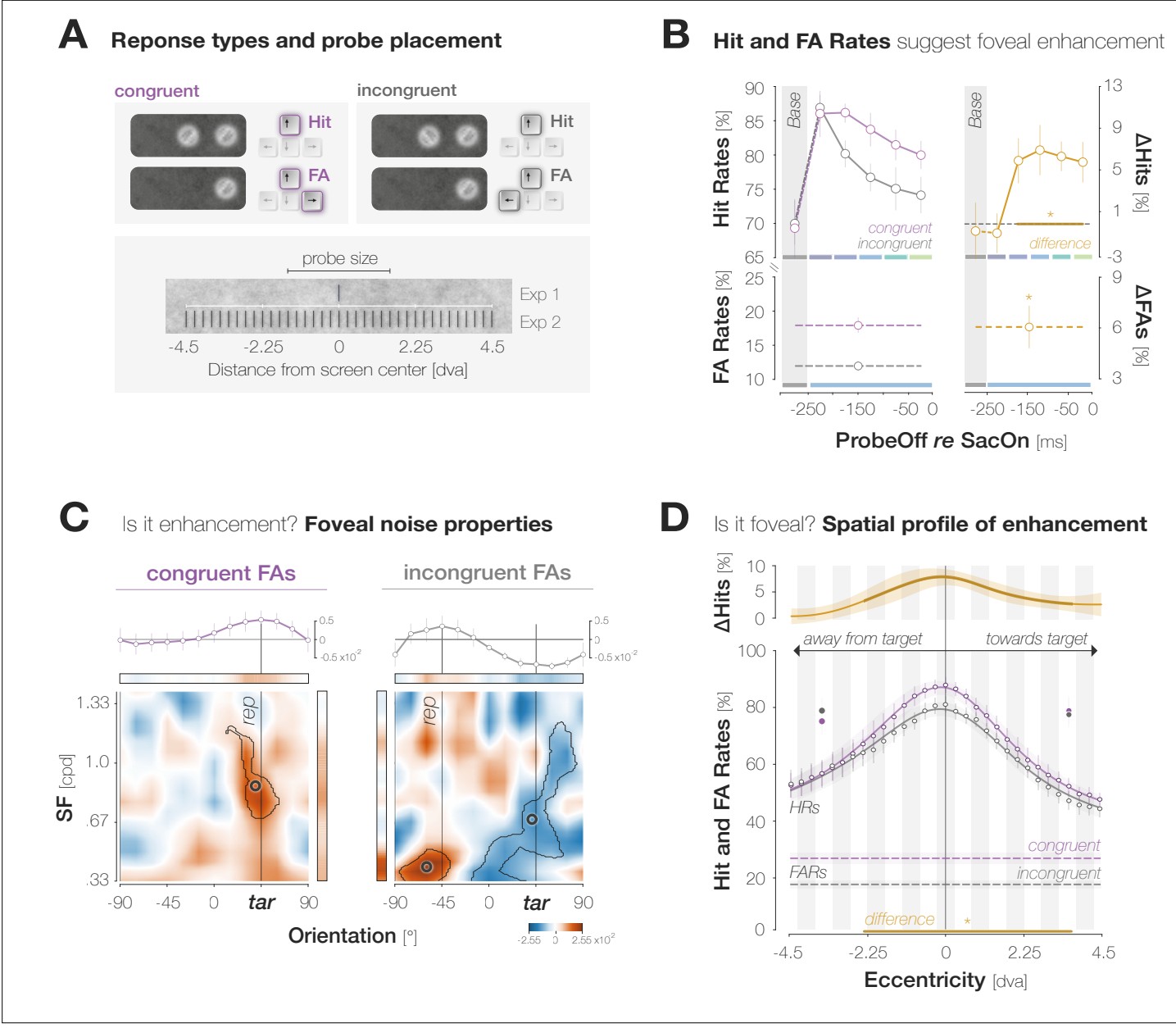

**Figure 2.** Evidence for predictive foveal enhancement of saccade target features. (**A**)*Top:* Response types. (In)congruent Hits are probe-present trials in which the probe was oriented (in)congruently to the target and observers reported 'present' (up arrow key). (In)congruent FAs are probe-absent trials in which observers reported the presence (up arrow key) of a probe with target-(in)congruent orientation (left, right arrow keys). *Bottom:* Probe placement. In Experiment 1, we chose a single probe location in the screen center. In Experiment 2, we defined 37 probe locations spaced evenly on a horizontal axis of 9 dva length around the screen center to measure the spatial profile of enhancement. (**B**) Hit and FA rates in Experiment 1 (y-axis) for different pre-saccadic time bins (x-axis; dark blue to light green rectangles), separated into congruent (purple) and incongruent (gray) responses. The difference between congruencies is plotted in brown. Asterisks indicate p-values ≤ .05 (obtained with bootstrapping, see Materials and methods). (**C**) Foveal noise properties on congruent and incongruent FA trials. We described the energy of 260 combinations of orientations (x-axis) and spatial frequency (SF; y-axis) in the foveal noise region (see Materials and methods). Noise images corresponding to trials with leftward target orientation were flipped, such that +45° corresponds to the target orientation ('tar'). The reported orientation ('rep') was either +45° (congruent FAs) or –45° (incongruent FAs). Orange regions indicate positive energy values in the standardized energy map (*z-score* >0) whereas blue regions indicate negative values (*z-score* <0). Marginal means along the orientation axis (curves and horizontal colored bars) are averages of energy values across all SFs. Marginal means along the SF axis (vertical colored bars) are averages of all SFs within the horizontal boundaries of the cluster around the reported orientation. Open circles indicate the center of mass of identified clusters. (**D**) Hit and FA rates in Experiment 2 (y-axis) for locations horizontally offset from the fovea (x-axis). Note that FAs cannot be spatially resolved. Data points indicate mean response rates across observers. The plotted curves are average Gaussian function fits to individual-observer means after aligning them to the mean recorded fixation position during saccade preparation (zero on the x-axis). Negative and

*Figure 2 continued on next page*

*Figure 2 continued*

positive x-axis values indicate probe locations away from and towards the saccade target, respectively. Thick brown lines highlight the significant portion of the spatial difference curve (congruent HRs – incongruent HRs; top panel; obtained with bootstrapping). Additional data points at ±3 dva eccentricity were measured by raising peripheral performance to a foveal level by adaptively increasing probe transparency (last session of Experiment 2). All error bands and bars denote ±1 SEM (n=7 in **B** and **C**; n=9 in **D**).

which the orientation of the probe had been reported correctly did not alter the nature of findings (*Appendix 1—figure 2*). Moreover, we observed a pronounced increase in overall HRs between the baseline and earliest pre-saccadic time point (69.7 ± 10.8% vs 86.5±6.7, p<0.001). This finding could be attributed to a predictive remapping of attention to the future retinotopic location of the saccade target – the fovea (*Rolfs et al., 2011*). Alternatively, observers may have approached the oculomotor and perceptual detection tasks sequentially. In early trial phases, they may have prioritized localizing the target and programming the eye movement. After motor planning had been initiated, cognitive resources may have been freed up for the foveal detection task. Yet another possibility is that the onset of the target stimulus created an urgency signal that transiently lowered decision thresholds (*Thura et al., 2012*). Crucially however, observers yielded similar incongruent HRs in the baseline and last pre-saccadic time bin (70.0 ± 8.8% vs 74.1 ± 7.0%, p=0.152, *BF10*=0.503). While we observed pronounced enhancement in the last pre-saccadic bin, congruent and incongruent HRs in the baseline bin were virtually identical. We therefore conclude that lower overall performance in the baseline bin did not prevent congruency effects from occurring. Instead, congruency effects started developing only after target appearance.

In sum, the orientation of the saccade target influenced observers' perceptual performance in their pre-saccadic center of gaze. While this suggests a predictive foveal enhancement of saccade target features, we substantiated two aspects of this assumption with further analyses and an additional experiment. First, though consistent with our hypotheses, a comparable increase in congruent HRs and FARs does not yield an advantage for congruent probes in classical sensitivity measures such as d-prime (*Green and Swets, 1966*). To ensure that FAs with target-congruent orientation report reflect an *enhancement* of orientation information in the foveal noise region, we investigated if those responses are FAs in the classical sense, that is, whether they reflect biased response behavior in our experiments. Alternatively, they may constitute a systematic reaction to incidental orientation information in the foveal noise region and therefore provide further support for a sensitization to target-congruent feature information.

## Analysis of foveal noise properties supports enhancement of foveal sensitivity

We separated all FA trials by whether observers had reported the target orientation (congruent FA) or the non-target orientation (incongruent FA). On each trial, we identified all noise images that had been displayed during the potential probe presentation period, that is, from the onset of the dynamic noise stream to saccade onset (for a conceptually similar approach using luminance-modulated patches see *Wilmott and Michel, 2021*). We described the properties of the foveal noise window in each image by determining the energy of 260 combinations of orientation and spatial frequency (ori*SF) at the potential probe location (Figure 6, Materials and methods). We subsequently identified lateralized clusters of high or low energy in the resulting maps using a combination of *t*-tests and bootstrapping procedures. We flipped the energy maps of trials in which the target was oriented to the left such that, in all subsequent analyses and plots,+45° corresponds to the target-congruent orientation while –45° corresponds to the incongruent orientation. Details are provided in the Materials and methods.

Energy maps underlying congruent and incongruent FAs showed a clear lateralization, with regions of high energy clustering around the perceived orientation (*Figure 2C*). Noise images associated with target-congruent FAs were characterized by a high energy around the target orientation. The identified cluster exhibited a center of mass (CM) close to +45° ($ori_{CM}$ = 39.38°, $SF_{CM}$ = 0.86 cycles per degree (cpd); tSum = 6625.7; p<0.001). Interestingly, this cluster included SFs from.67 cpd to 1.19 cpd – a range that covers the pre-saccadic development of visual resolution at the target of a 10 dva saccade (*Kroell and Rolfs, 2021*). This suggests that information about the target was indeed available for foveal processing, at a resolution influenced by the target's pre-saccadic eccentricity. Feedback information may have interacted with foveal noise content, allowing those ori*SF combinations

that are congruent with the fed-back signal – but not others – to drive response behavior. Indeed, separating noise images that had appeared before or after target onset revealed that the cluster encompassing higher SFs manifested exclusively during saccade preparation ($ori_{CM}$ = 40.31°, $SF_{CM}$ = 0.57 cpd; tSum = 22,099.0; p<0.001; *Appendix 1—figure 3*).

Noise images associated with target-incongruent FAs exhibited a high energy around the perceived, non-target orientation. The corresponding cluster was slightly repelled from the target orientation ($ori_{CM}$ = 60.0°) and confined to very low SFs from.33 to.51 cpd ($SF_{CM}$ = 0.41 cpd; tSum = 5587.7; p =< 0.001). These SFs likely generated a salient percept when flashing on screen, motivating observers to report the non-target orientation in a purely stimulus-driven fashion. Unlike congruent FAs, incongruent FAs were not associated with a higher-SF cluster around the perceived orientation. Instead, the underlying noise images exhibited an absence of evidence for the target orientation that manifested in a cluster of low energy centered on 41.25° ($SF_{CM}$ = .68 cpd; tSum = –20,050; p=0.002). Whereas the low-SF portion of this cluster relied on images visible in the baseline time bin, the portion encompassing target-like orientations in a higher SF spectrum manifested exclusively during saccade preparation (*Appendix 1—figure 3*). Combined, the high-energy cluster repelled from the target orientation, and the low-energy cluster covering a wide SF range around the target orientation suggest that throughout the trial, noise content as target-dissimilar as possible was required for observers to perceive a competing orientation in the foveal noise. Finally, we compared the amount of evidence required to perceive a certain orientation by summarizing the absolute filter responses within the identified clusters for each response type. Since incongruent FAs required both, perceptual evidence for the non-target orientation and an absence of evidence for the target orientation, the total amount of evidence necessary to trigger incongruent FAs exceeded the evidence underlying congruent FAs by a factor of 3.4, p<0.001.

To summarize, just like Hits, FAs constitute a systematic reaction to foveal visual input. Rather than classical FAs, these responses therefore seem to constitute Hits triggered by stimulus information with a lower signal-to-noise ratio. Congruency effects in FARs can be interpreted in the same way as congruency effects in HRs: information about the peripheral saccade target interacts with congruent foveal input and facilitates its detection.

## Enhancement is spatially confined to the foveal region

The congruency effects described above may indeed reflect a spatially specific interaction between peripheral and foveal input. Alternatively, feature-based attention to the target orientation may yield a widespread enhancement of congruent orientations which could encompass the foveal region without being confined to it. To test if predictive enhancement is restricted to the pre-saccadic center of gaze and its immediate vicinity, we conducted a second experiment in which the probe, if presented, appeared in one of 37 locations spaced evenly on a horizontal axis from 4.5 dva to the left to 4.5 dva to the right of the screen center (in increments of 7–8 pixels). On each trial, the position of the probe was unpredictable for the observer. We determined congruent and incongruent HRs within a moving window including 6 adjacent locations and described the resulting spatial profiles by fitting Gaussian curves with position-invariant vertical offsets to individual observer data (*Figure 2D*; see Materials and methods for details and *Appendix 1—figures 4–5* for alternative fits). We flipped trials with leftward saccades, such that negative and positive position values indicate probe locations away from and towards the saccade target, respectively. To account for small fixation errors, spatial profiles are aligned to the mean fixation position during the saccade preparation period.

Across congruencies, HRs were highest in the center of gaze and decreased continuously as the eccentricity of the probe increased. While feature-based attention would predict a uniform detection advantage for congruent probes across all tested locations, the difference between the curves, and therefore enhancement, was most pronounced in the center of gaze ($HR_{cong-incong}$=7.7 ± 4.8% at –0.1±1.7 dva; p<0.001). Enhancement remained significant within a region of 6.4 dva around the pre-saccadic fixation, extending further towards the saccade target than away from it: congruent HRs significantly exceeded incongruent ones from –2.6 to +3.8 dva. The congruent HR profile exhibited a significantly higher peak than the incongruent one (80.4 ± 7.8% vs 87.9 ± 7.3%, p<0.001, *BF10*=22.689). At the same time, and in opposition to a global enhancement of target-congruent orientations, the vertical offsets of the congruent and incongruent profile did not differ significantly (39.3 ± 19.2% vs 35.9 ± 16.3%, p=0.323, *BF10*=0.340).

Note that the size of the stimulus used to probe performance will presumably determine the width of the enhancement profile. Since the probe stimulus in our experiment had a comparably large radius of 1.5 dva, the leftmost significant point at –2.6 dva could indicate enhancement of a probe centered on –1.1 dva (and thus covering the very center of gaze). Consequently, enhancement may be confined to an even narrower region than our data indicate. While the width of the enhancement profile should be interpreted with caution, the main conclusions that can be drawn are that enhancement (*i*) peaks in the center of gaze, (*ii*) is not uniform throughout the tested spatial range as, for instance, global feature-based attention would predict, and (*iii*) is asymmetrical, extending further towards the saccade target than away from it. We additionally inspected the influence of saccadic precision on the width of the enhancement profile. This analysis did not alter our conclusions and is presented in the Supplements (*Appendix 1—figure 6*).

Again, congruent FARs significantly exceeded incongruent ones (FAR$_{cong-incong}$=9.4 ± 7.3%; p<0.001). An inspection of noise properties revealed that – despite observers' explicit knowledge about the range of probe locations – FAs were primarily triggered by foveal orientation information (Supplements; *Appendix 1—figure 7* and *Appendix 1—Video 1*).

## The lack of parafoveal enhancement is not a consequence of low performance at larger eccentricities

Enhancement may depend on the baseline performance level at a given eccentricity. In other words, sensitivity for parafoveal probes may simply be too low for congruency effects to emerge. To ensure that an absence of enhancement for more eccentric probes is a true consequence of their location and not caused by a simple, eccentricity-related decrease in visual performance, observers completed an additional session in which the probe appeared either 3 dva to the left or 3 dva to the right of the screen center. Crucially, we administered a staircase procedure to determine the probe contrast at which performance for *parafoveal* target-incongruent probes would be just as high as *foveal* performance had been in the preceding sessions. This manipulation was successful: incongruent HRs for parafoveal probes were statistically indistinguishable from incongruent HRs for probes presented in the center of gaze (–3 dva: 79.1±14.8% vs 81.3 ± 8.4%, p=0.489, *BF10*=0.400; 3 dva: 77.7±15.5% vs 81.3 ± 8.4%, p=0.415, *BF10*=0.435). Even though performance had been raised to a foveal level, congruent and incongruent HRs differed neither for probes appearing away from (HR$_{cong-incong}$=−3.7 ± 10.0%, p=0.127, *BF10*=0.531) nor towards (HR$_{cong-incong}$=1.4 ± 9.0%, p=0.336, *BF10*=0.352) the saccade target (see individual data points at ±3 dva in *Figure 2D*).

## Enhancement is aligned to the center of gaze, not to the remapped target location

Conceptually, the pre-saccadic center of gaze corresponds to the predictively remapped location of the saccade target (*Rolfs et al., 2011*; *Collins et al., 2009*; *Wilmott and Michel, 2021*). Enhancement may thus be aligned to the remapped target location rather than to the center of gaze per se (*Knapen et al., 2016*). To distinguish between both possibilities, we took advantage of the observation that the precise remapping vector depends on saccade endpoints on an individual-trial level (*Collins et al., 2009*): for hypometric horizontal saccades, the predictively remapped target location is shifted into the hemifield of the target by the magnitude of the undershoot (*Figure 3C*, top panel). Conversely, for hypermetric horizontal saccades, the predictively remapped target location is shifted into the opposite hemifield by the magnitude of the overshoot. Following this logic, we aligned probe locations to either the recorded pre-saccadic center of gaze or the predictively remapped target location on an individual-trial level. Since the spatial offset between the center of gaze and the remapped target location increases with saccadic landing error, we separated all recorded eye movements into 'accurate' (the saccade landed on the target stimulus, that is, no more than 1.5 dva from its center) and 'inaccurate' (the saccade landed off-target but within the accepted landing radius of 3.25 dva around it). Note that we increased the width of the moving boxcar window to nine locations in order to guarantee sufficient trial numbers in each condition.

When probe locations were aligned to the center of gaze, considering only accurate saccades yielded significant enhancement from –2.6 to 2.1 dva and from 3.2 dva throughout the measured range towards the saccade target (*Figure 3A*). Inaccurate saccades showed a more pronounced asymmetry (*Figure 3B*): enhancement reached significance between –1.1 and 4.4 dva. An increased

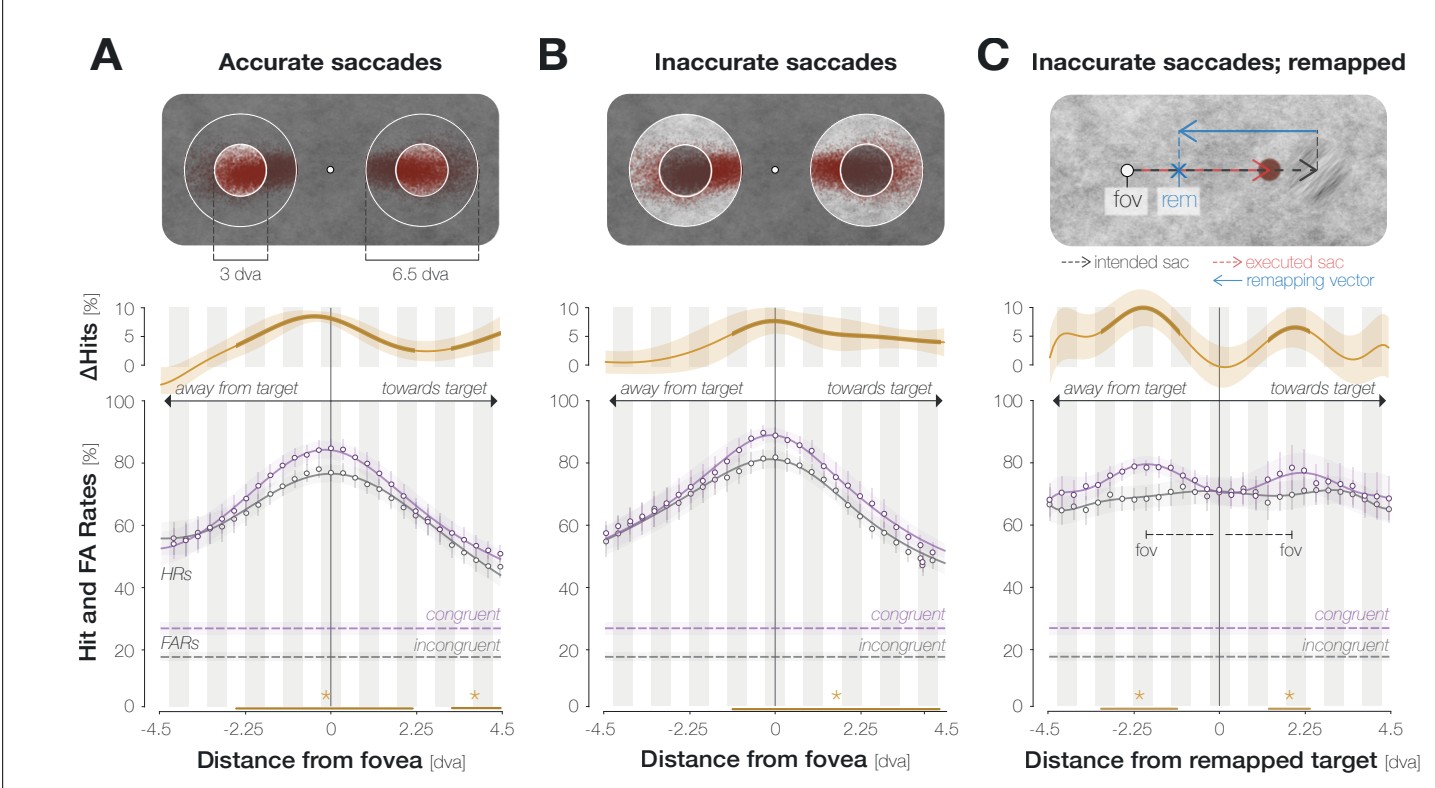

**Figure 3.** Enhancement is aligned to the center of gaze, not to the remapped target location. (**A**) Spatial HR profiles for the subset of trials in which the saccade landed on the target ('accurate'; see top panel). (**B**) Spatial profiles for saccades that landed off-target but within the accepted landing area ('inaccurate'; see top panel). In (**A**) and (**B**), zero on the x-axis corresponds to the recorded pre-saccadic center of gaze. The top panels illustrate the endpoints of all included saccades separately for leftwards and rightwards eye movements. (**C**) Spatial profiles for inaccurate saccades when probe locations were aligned to the remapped target location rather than to the center of gaze. The top panel illustrates a trial in which an observer is fixating on the fixation dot ('fov') and planning a saccade to the center of the target stimulus ('intended sac'; black dashed arrow). The observer executes a hypometric saccade (red arrow), the endpoint of which is illustrated as a red disk. The remapping vector (blue arrow) is obtained by flipping the executed saccade vector and attaching it to the intended saccade endpoint, i.e., to the center of the target (*Collins et al., 2009*). In all spatial profiles in (**C**), zero on the x-axis indicates the remapped target location (blue X), that is, the location that the saccade target would occupy upon saccade landing. With this alignment, we observed two peripheral maxima that correspond to the mean center of gaze ('fov'). All conventions for the spatial profiles follow those of *Figure 2D*. Remapping-aligned curves for all trials and accurate saccades only are provided in the Supplements (*Appendix 1—figure 8*).

asymmetry for inaccurate saccades may indeed be related to predictive remapping: since inaccurate saccades were hypometric on average (Mdn = 9.7 ± 0.4 dva), asymmetric enhancement would have boosted congruency at the remapped target location across all trials. Yet, aligning probe positions to the remapped target location on an individual-trial level did not lead to a narrowing of the enhancement profile (*Figure 3C*). Instead, we observed two peripheral maxima. Their location on the x-axis equals the mean remapping-dependent leftwards (2.0±0.4 dva) and rightwards (1.9±0.3 dva) displacement across trials. In other words, they correspond to the center of gaze.

In sum, the enhancement profile is more asymmetric for inaccurate eye movements. An increase in asymmetry could bear functional advantages since it would boost congruency at the remapped target location across all trials. Importantly though, this adjustment seems to rely on an estimate of average rather than single-trial saccade characteristics: aligning probe locations to the remapped attentional locus on an individual trial level provides further evidence that, irrespective of individual saccade endpoints, enhancement was aligned to the fovea.

## Post-saccadic target foveation boosts congruency

To investigate if post-saccadic target foveation influences pre-saccadic detection judgments, we removed the target during saccadic flight on half of the trials in Experiment 1 (*Pre*). On the remaining trials, the target remained visible for a brief duration after saccade offset (*Pre-Post; Mdn = 22.3 ±*

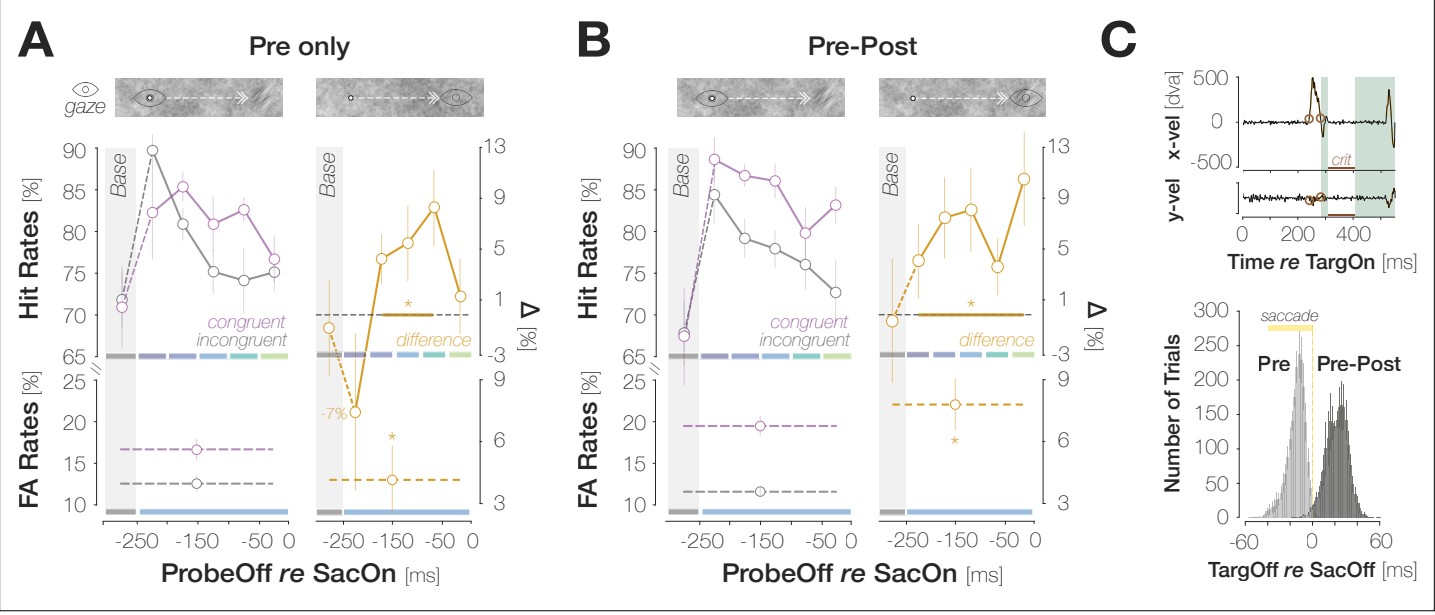

**Figure 4.** Post-saccadic target foveation boosts congruency. (**A**) Foveal Hit Rates (top row) and FA Rates (bottom row) for the subset of trials in which the saccade target had disappeared during saccadic flight ('Pre-only'). (**B**) Foveal Hit Rates (top row) and FA Rates (bottom row) for the subset of trials in which the saccade target had remained visible briefly after saccade landing (right column; 'Pre-Post'). Conventions in (**A**) and (**B**) are as in *Figure 2B*. (**C**) *Top*: Illustration of saccade inclusion criteria based on horizontal and vertical eye velocities (x vel; y vel) recorded in an example trial. Orange outlines highlight offline-detected saccadic events. To ensure that foveal input remained stable on the retina after landing, we excluded trials in which a second saccade occurred in a critical time window ('crit') between 25 and 100ms after response saccade offset. We employed a conservative estimate of saccade offset and excluded post-saccadic oscillations from the saccadic profile. We therefore introduced a short time window of 25ms after response saccade offset during which saccadic activity did not lead to trial exclusions. *Bottom*: Histogram of time intervals between target offset and saccade offset. On *Pre* trials, the target disappeared ~14.1ms (median) before saccade landing. On *Pre-Post* trials, the target remained visible for ~22.3ms (median) after saccade landing.

4.12ms). While we observed significant congruency effects when the target was visible exclusively during saccade preparation, congruency was more pronounced in *Pre-Post* trials (*Figure 4*): across all pre-saccadic time bins, the difference in HRs amounted to 2.5 ± 6.0% in *Pre* and to 6.9 ± 2.8% in *Pre-Post* trials, p<0.001. Target foveation impacted the time course of congruency effects: when the target disappeared during the saccade, congruent HRs significantly exceeded incongruent ones in medium stages of saccade preparation ([–250 –200] ms: $HR_{cong-incong}$=4.4 ± 4.9%; p=0.003; [–200 –150] ms: $HR_{cong-incong}$=5.6 ± 7.7%; p=0.024; [–150 –100] ms: $HR_{cong-incong}$=8.4 ± 7.6%; p<0.001). Congruent and incongruent HRs in the earliest and latest time bins did not differ significantly ($HR_{cong-incong}$ = –7.4 ± 1.6%; 1.5 ± 7.6%; ps = 0.060; 0.283). In *Pre-Post* trials, congruent HRs numerically exceeded incongruent HRs throughout saccade preparation, though this difference failed to reach significance in the earliest bin ($HR_{cong-incong}$=5.6 ± 7.5%; p=0.055). In all later bins, congruent HRs significantly exceeded incongruent HRs ($HR_{cong-incong}$=7.6 ± 8.3%, 8.2 ± 8.3%, 3.8 ± 5.7%, 10.5 ± 9.5%; all ps <0.041). A brief foveation of the target was sufficient to boost congruency. We determined the difference between congruent and incongruent HRs within a boxcar window (width: 10ms; step size: 1ms) sliding across the range of post-saccadic target durations (1–48ms). Congruency was most pronounced in the window from 11 to 21ms ($HR_{cong-incong}$=9.1 ± 3.4%). In the baseline bin, congruent and incongruent HRs were virtually identical for both *Pre* ($HR_{cong-incong}$ = –0.9 ± 9.8%, p=0.534) and *Pre-Post* ($HR_{cong-incong}$ = –0.4 ± 12.6%, p=0.590) trials.

## Pre-saccadic congruency effects are more pronounced and faster than congruency effects during fixation

Our findings suggest that foveal feedback is effective while an eye movement is prepared to a different visual field location. We conducted a third experiment to compare the magnitude and time course of pre-saccadic congruency effects from Experiment 1 to congruency effects during fixation. Observers performed the identical foveal detection task but maintained fixation in the screen center throughout

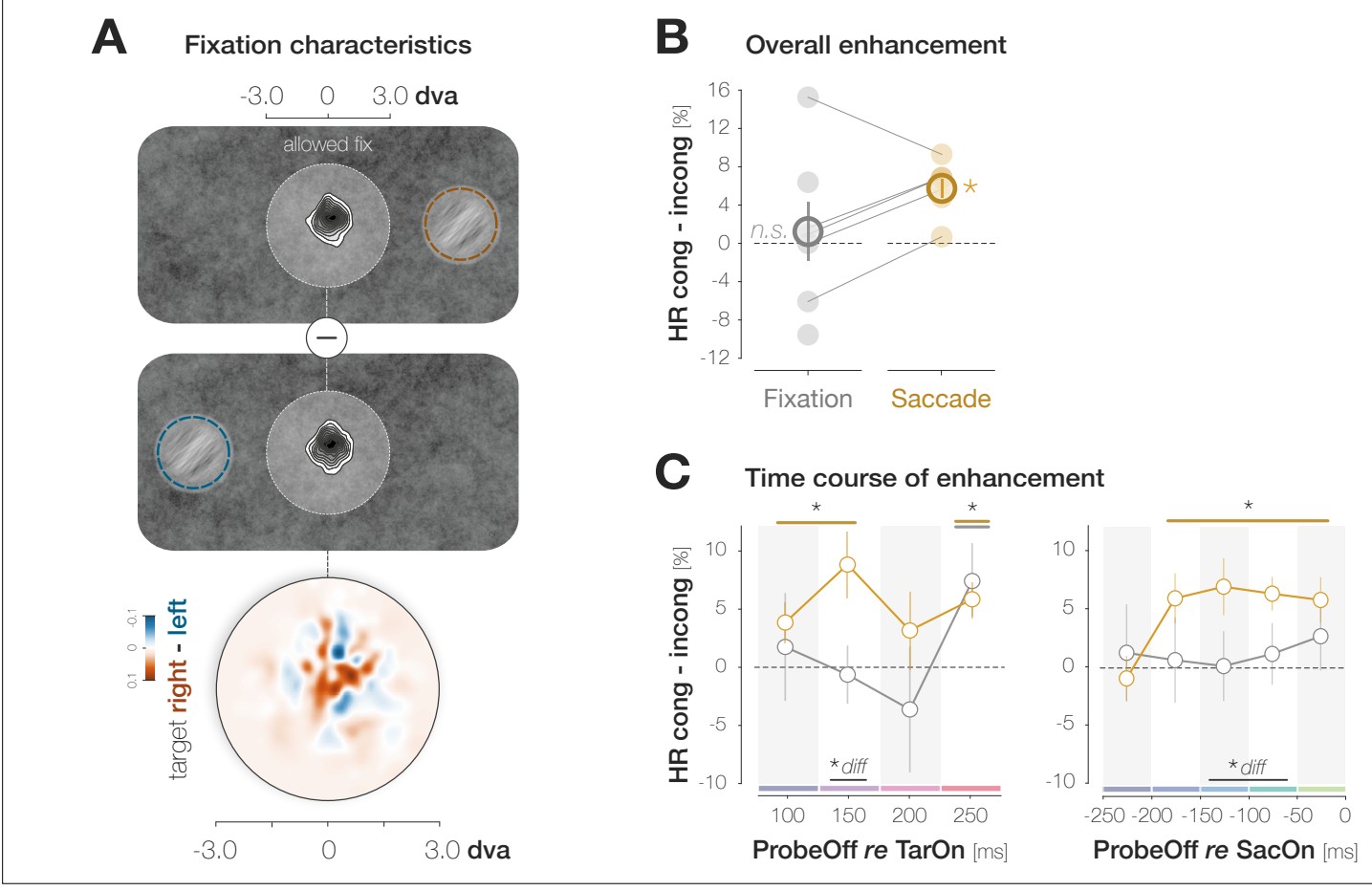

**Figure 5.** Saccade preparation accelerates foveal enhancement. (**A**) Observers' mean fixation densities when the target was presented to the right (first panel; orange) or the left (second panel; blue) of the screen center. The difference plot (third panel) reveals no systematic biases in gaze direction. The distance between the allowed fixation area around the screen center (light disk) and the target is reduced for illustration purposes. (**B**) Mean enhancement across all target-probe presentation intervals for the fixation (gray symbols) and saccade experiment (Experiment 1; orange symbols). Saccade data were collected in Experiment 1. Filled disks correspond to individual observers. Lines connect the data points of observers who participated in both the saccade and fixation experiment (n=5). (**C**) Time course of enhancement locked to target onset (left) or saccade onset (right). As in (**B**) orange symbols correspond to saccade data while gray symbols correspond to fixation data. The orange curve in the second panel is identical to the difference curve plotted in *Figure 2B*. Across panels, error bars denote ±1 SEM (n=7 for saccade and fixation data, respectively). Horizontal lines and asterisks above the data points highlight comparisons to zero with a p ≤ 0.05 for the fixation (gray) and saccade (orange) condition (obtained with bootstrapping). Horizontal lines and asterisks below the data points highlight bins in which the saccade and fixation condition differed significantly ('diff').

the trial. After excluding trials in which observers had executed a (micro-) saccade (3.0% of trials), we inspected the fixation densities on all remaining trials for small-scale gaze biases towards the target location (*Figure 5A*). Since we did not observe a lateralization of fixation densities, congruency effects – if present – are unlikely to be caused by small-scale oculomotor activity. To gauge the magnitude of enhancement, we computed the mean congruency effect across all probe presentation time points in the fixation condition. In contrast to the mean pre-saccadic congruency effect from Experiment 1, the mean congruency effect during fixation failed to reach significance (1.5 ± 4.7%, p=0.277, *BF10*=0.378; *Figure 5B*). Five of the seven observers tested in the fixation condition had participated in Experiment 1. We matched the saccade and fixation data of those participants to compute comparisons on a within-observer level. Across all probe presentation time points, the magnitude of the pre-saccadic congruency effect significantly exceeded the mean congruency effect during fixation, p=0.046. To compare the target-locked time course between fixation and saccade preparation, we first aligned congruency effects from Experiment 1 to the onset of the target rather than to saccade onset (*Figure 5C*; left). Pre-saccadic congruency effects emerged rapidly: they reached significance when

the probe was presented 100ms after the target (3.9 ± 4.7%, p=0.008) and peaked at a target-probe interval of 150ms (8.8 ± 7.6%, p<0.001). Congruency effects during fixation, in turn, developed more slowly and only reached significance when the probe appeared 250ms after the target (7.4 ± 8.6%, p=0.007; all remaining time points: ps >0.379; *BF10*s<0.425). To reconstruct a movement-locked time course in the fixation condition, we determined the proportion of each target-probe presentation interval (50, 100, 150, and 200ms) in every pre-saccadic time bin in Experiment 1. We then computed the inner product of the target-locked congruency effects during fixation (*Figure 5C*, first panel) and the proportion of all four target-probe intervals in a specific pre-saccadic time bin (*Rolfs and Carrasco, 2012*). In other words, we equated the contribution of different target-probe presentation intervals to the fixation and saccade condition. As a consequence, all differences between fixation and saccade data in the second panel of *Figure 5C* can be attributed to the preparation of an eye movement. While congruency effects in the saccade condition reached significance from the second ([–200 –150[ ms) to the last ([–50 0[ ms) pre-saccadic time bin (all ps <0.003), congruency effects in the fixation condition failed to reach significance throughout (all ps >0.16).

## Discussion

We demonstrate that defining features of an eye movement target are predictively enhanced in the pre-saccadic center of gaze. Foveal enhancement was both temporally and spatially specific, more pronounced when the saccade target was foveated briefly after saccade landing and accelerated during saccade preparation as compared to fixation.

### Is the fovea special? Potential mechanisms of foveal prediction

Enhancement was spatially specific, that is, most pronounced in the center of gaze and confined to an area of 6.4 dva around it. As noted above, the true profile of enhancement may be even narrower: since our probe exhibited a comparably large diameter of 3 dva, congruency effects at eccentric locations may rely at least partly on the facilitated detection of the probe's near-foveal margins. Could congruency effects with a similar spatial profile build up around any other relevant and therefore attended location in the visual field? The feasibility of peripheral congruency effects depends on the mechanism assumed to underlie our findings:

### Joint modulation of spatial and feature-based attention

Foveal enhancement may rely on a joint modulation of spatial and feature-based attention (*White et al., 2015*; *Ibos and Freedman, 2016*). Saccade preparation entails the emergence of two spatially confined attention pointers (*Cavanagh et al., 2010*): one centered on the saccade target and one centered on its predictively remapped location, that is, the foveal region (*Rolfs et al., 2011*; *Collins et al., 2009*). At the same time, the appearance of the saccade target in our investigation may have introduced global feature-based attention to its orientation (*White and Carrasco, 2011*). The combination of spatial attention pointers carrying no feature information, and feature-based attention lacking spatial tuning may indeed achieve what we observe: a spatially specific alteration of visual sensitivity to defining features of a stimulus presented elsewhere in the visual field. It is conceivable that the fovea as the region of highest acuity is assigned a permanent attention pointer. In principle, however, this mechanism could operate across the visual field and may underlie previous findings that demonstrate an interaction of feature information between peripheral locations [for motion discrimination (*Szinte et al., 2016*), crowding (*Harrison et al., 2013*), and adaptation (*He et al., 2018*; *Biber and Ilg, 2011*; *Melcher, 2007*)].

Two spatially distinct, feature-selective attention pointers may account for the impact of post-saccadic target presence on the pre-saccadic time course of congruency effects in our investigation. Observers likely reported the presence of a target-congruent foveal probe whenever perceptual evidence for its orientation had exceeded a certain threshold. Assuming that orientation information was sampled simultaneously from the foveal and peripheral attentional focus, and assuming that the foci persisted for a brief period after saccade landing (*Golomb, 2019*; *Jonikaitis et al., 2013*), the salient post-saccadic foveal view of the target may have allowed even early and therefore subthreshold pre-saccadic foveal information to contribute to an above-threshold signal.

Nevertheless, this mechanism falls short of accounting for some aspects of our own as well as previous findings. First, we did not observe congruency effects when the probe appeared in one of two possible parafoveal locations to which spatial attention pointers could have been allocated strategically. Second, congruency effects were aligned to the center of gaze rather than to the precise, predictively remapped location of the target, the coordinates of which depend on saccade endpoints on an individual-trial level (*Collins et al., 2009*). Third, foveal retinotopic cortex contributes to the processing of complex peripheral shapes (*Williams et al., 2008*; *Fan et al., 2016*; *Chambers et al., 2013*; *Yu and Shim, 2016*; *Weldon et al., 2020*). Whether feature-based attention can operate at this level of abstraction, and whether the proposed mechanism is viable for naturalistic objects involving feature conjunctions, is unclear. Consequently, the fovea may not merely be one of many locations an attention pointer can be allocated to. Instead, its unique characteristics seem to be harnessed in a way that would not be viable at other visual field locations.

## Feedback connections to foveal neurons

Visual processing does not operate in a strictly feedforward fashion (*Rockland and Van Hoesen, 1994*; *Bullier, 2001*). For instance, neurons in several temporal areas (e.g. TE, IT, TPO, STS), most of which are associated with the computation of position-invariant object information, project back to primary visual cortex and potentially relay feature information about relevant peripheral stimuli to V1 neurons with foveal receptive fields (*Rockland and Van Hoesen, 1994*; *Bullier, 2001*). Crucially, peripheral features can be decoded from brain activation in foveal – but not other peripheral – retinotopic areas (*Williams et al., 2008*), supporting the fovea's singular role as an active blackboard. Moreover, encoding models based on deep convolutional neural networks used to predict neural responses to natural scene stimuli in a purely bottom-up fashion consistently fail to account for foveal activity in visual cortical areas (*Mell et al., 2021*). Foveal feedback connections, however, could account for all aforementioned findings: irrespective of the precise remapping vector, and despite the possibility to allocate peripheral attention pointers, feature information would be invariably relayed to the fovea. Moreover, temporal areas encode complex shapes, a coarse sketch of which could be fed back to foveal retinotopic cortex (*Bullier, 2001*). An involvement of foveal feedback connections in saccade preparation appears physiologically feasible: neurons in the previously mentioned temporal areas exhibit median response latencies of 50–130 ms (*Bullier, 2001*). Feedback delays to foveal retinotopic cortex would thus lie well within the range of typical saccade latencies (*Becker, 1972*). Our results are consistent with this timing: we observed maximum enhancement when the foveal probe appeared 150–200ms after the target.

## Is saccade preparation special?

While a link between foveal feedback and saccade preparation has been suggested repeatedly (*Fan et al., 2016*; *Chambers et al., 2013*; *Yu and Shim, 2016*), the influence of foveal processing on peripheral task performance has been studied almost exclusively during passive fixation. The only study investigating foveal feedback in an active setting revealed that a foveal distractor no longer impacted peripheral discrimination performance when observers prepared a saccade *away* from the to-be discriminated object (*Fan et al., 2016*). These findings support the main assumption motivating the current study: saccades automatically establish a transient connection between the current and future foveal location, that is, the saccade target and the pre-saccadic center of gaze. A similar connection between foveal and peripheral input may exist or be inducible by task demands during fixation. Arguably though, this connection is strengthened considerably when a saccade successively projects two otherwise unrelated visual field locations (the current center of gaze and the saccade target) onto the same retinal location (the fovea).

Natural visual environments are crowded and, more often than not, contain multiple objects at a time. If foveal congruency effects rely on feedback connections, some selection mechanism is required that determines which object or object feature is processed in temporal areas and subsequently fed back to foveal cortex. During saccade preparation, this selection mechanism would emerge naturally: pre-saccadic attention shifts to the eye movement target in an obligatory fashion (*Deubel and Schneider, 1996*; *Moore and Fallah, 2001*) and transiently prioritizes the target over any other location in the visual field. Pre-saccadic attention shifts faster than covert attention does (*Rolfs and Carrasco, 2012*) and even uncrowds the saccade target from surrounding objects (*Harrison et al.,*

*2013*). Most importantly, pre-saccadic attention is a natural relevancy filter – in spontaneous visual behavior, gaze is directed to the currently most significant object in the scene (*Einhäuser et al., 2008*; *Nuthmann and Henderson, 2010*; *'t Hart et al., 2013*). All these properties identify pre-saccadic attention as a parsimonious and naturally emerging selection mechanism that may even be potent enough to shape feedforward processing: at the moment the feedback mechanism is engaged, most of visual cortex may indeed be occupied with the processing of saccade target features. Sufficient selection can likely be achieved through purely covert attention in the absence of eye movements. Nonetheless, feedback effects are expected to emerge more slowly (*Rolfs and Carrasco, 2012*). This is indeed what we observe.

It has been suggested that foveal feedback during fixation reflects the automatic preparation of an eye movement that is simply not executed (*Fan et al., 2016*; *Chambers et al., 2013*; *Yu and Shim, 2016*). In this case, feedback effects would emerge during fixation inasmuch as the respective task engages the saccadic system. Our paradigm likely covers the conservative end of this spectrum: we intentionally made the features of the target as task-irrelevant as possible, contrary to previous investigations. We intended to ensure that potential congruency effects would be automatic and not induced by a peripheral discrimination task. Moreover, the target was smoothly embedded in background noise and presented at a medium opacity and a comparably large eccentricity of 10 dva. It is conceivable that increasing the conspicuity of the target by reducing its eccentricity or transparency against the background noise would trigger the saccadic system to a larger extent and thereby boost congruency effects during fixation. In general, strong evidence that foveal prediction serves saccade preparation and, due to existing neuronal connections, spills over to any fixation task would be provided if the oculomotor characteristics of individual observers (such as their typical saccade latency) influence properties of the foveal congruency effect (such as its time course) even during passive fixation (*Rolfs and Schweitzer, 2022*).

## Can our findings be explained by established mechanisms other than foveal prediction?

Since enhancement was aligned to the center of gaze irrespective of individual saccade endpoints, we conclude that our findings are not a simple correlate of predictive remapping (*Collins et al., 2009*). Moreover, pre-saccadic foveal enhancement was more pronounced than its equivalent during fixation, suggesting that bottom-up covert attention shifts cannot account for our results. For similar reasons, foveal enhancement does not seem to be a mere consequence of sensory information accumulation: while the presentation duration of the target in the fixation condition was equal to the presentation duration of the saccade target in Experiments 1 and 2, congruency effects differed markedly. Furthermore, enhancement did not increase monotonically across the target presentation period. Especially in the PRE-only condition (*Figure 4A*), congruency effects were most pronounced in medium stages of saccade preparation, in accordance with established neuronal feedback latencies, and vanished in later stages.

Similar arguments support the notion that our findings reflect perceptual enhancement rather than a shift in decision criterion. A criterion shift would have affected HRs and FARs alike. Nonetheless, the difference in HRs manifested under specific and meaningful conditions: pre-saccadic enhancement was temporally specific – congruent and incongruent HRs were virtually identical when the probe appeared in a baseline bin or during early saccade preparation (*Figure 2B* and *Figure 4A*) and only started to differ later. Crucially, the time course of enhancement during saccade preparation and fixation mirrored the typical temporal development of visual sensitivity during pre-saccadic attention shifts and covert attentional allocation, respectively (*Li et al., 2021*; *Rolfs and Carrasco, 2012*). We are unaware of literature demonstrating comparable temporal specificity for a shift in decision criterion. It has furthermore been argued that a defining feature of innately perceptual effects is their spatial specificity whereas criterion shifts should manifest in a spatially global fashion (*Fritsche et al., 2017*). In our investigation, congruent HRs exceeded incongruent ones within a confined spatial region around the center of gaze. We did not observe enhancement for probes presented at ±3 dva eccentricity even when we raised parafoveal performance to a foveal level by adaptively increasing the opacity of the probe. The average saccadic accuracy or, more specifically, the mean remapped target location influenced the spatial asymmetry of enhancement (*Figure 3B*), in a fashion that is reconcilable with previous findings (*Collins et al., 2009*). Nonetheless, we would like to mention that

a signal detection model assuming location-specific variations in choice criterion could account for behavioral data recorded in chickens and macaque monkeys (*Sridharan et al., 2017*). Model-based simulations suggest that attentional effects commonly interpreted as changes in perceptual sensitivity could be explained equally well by assuming localized variations of criterion. A decisive difference between the nature of criterion shifts in these simulations and our findings is that, while the simulated criterion changes would affect any visual information presented at a certain visual field location, our results would require an interplay between spatially specific and feature specific criterion shifts. Only a criterion shift that selectively affects orientation information which appears in or near the center of gaze and, crucially, matches the orientation of the peripheral saccade target could account for our findings. Variations in criterion that are temporally, spatially and feature selective, follow the time course of pre-saccadic or covert attention depending on observers' oculomotor behavior, do not remain effective throughout an entire trial, are sensitive to the mean remapped target location across trials, and do not affect parafoveal stimuli even after their opacity has been increased to match foveal performance would be unprecedented in the literature and, even if existent, appear just as function-ally meaningful as sensitivity changes occurring under the same conditions.

Similarly, we consider it unlikely that the overall HR time course relies on a temporal shift in crite-rion. More specifically, the continuous decrease in congruent and incongruent HRs across the saccade preparation period may reflect a gradually increasing decision criterion rather than a decrease in foveal visual sensitivity. Yet, our results are in line with the finding that, compared to fixation, orienta-tion discrimination performance during saccade preparation decreases in the fovea as it increases at the saccade target location (*Hanning and Deubel, 2022*). In this investigation, observers executed saccades towards unfiltered noise while discriminating the orientation of a brief signal presented at an unpredictable location (2AFC). Since both the orientation and location of the probe stimulus were fully randomized, systematic shifts in criterion for a certain feature, location or combination of both are unable to account for these results. A parsimonious explanation of these findings and our data pattern is therefore that pre-saccadic foveal sensitivity decreases as attention shifts from the current center of gaze to the saccade target.

Lastly, we would like to share a phenomenological impression that was spontaneously reported by 6 out of 7 observers in Experiment 1 and experienced by the author L.M.K. many times. On a small subset of trials, participants in our paradigms have the strong impression of *perceiving* the target in the pre-saccadic center of gaze. This percept is rare but so pronounced that some observers inter-rupted the experiment to ask which probe orientation they should report in case they had detected two on the same trial. Interestingly, the actual saccade target and its foveal equivalent are perceived simultaneously in different spatiotopic locations, suggesting that this percept cannot be ascribed to a temporal misjudgment of saccade execution (after which the target would have actually been fove-ated). We have no data to prove this observation but would nonetheless like to share it. Experiencing it ourselves has left us with no doubt that the fed-back signal is truly – and almost eerily – perceptual in nature.

## Does foveal prediction transfer to other visual features and complex natural environments?

While we characterized foveal prediction for orientation information, it would be interesting to examine whether and to what extent similar effects could be observed for various stimulus properties. On the one hand, foveal processing is optimized for surface features such as orientation, SF and color, and may be recruited for the peripheral processing of primarily those properties. On the other hand, and during saccade preparation in particular, any visual information at the eye movement target will be processed foveally upon landing. In consequence, the prediction of all visual features, even those for which foveal processing bares no significant advantage over peripheral processing (such as tempo-rally modulated signals, *McKee and Nakayama, 1984*), appears adaptive. Coherent motion at the saccade target, for instance, causes immediate ocular following upon saccade landing (*Kwon et al., 2019*), a response that may well rely on foveal prediction.

In naturalistic environments, visual information at both the saccade target and foveal location will most often be characterized by feature conjunctions. The feedback mechanism itself seems capable of operating at this complexity: high-level peripheral information such as object category has been successfully decoded from foveal neuronal activation (*Williams et al., 2008*). Nonetheless, how

fed-back information interacts with high-contrast feedforward foveal input – as it is likely required to when gaze is spontaneously directed from one object of interest to another – remains to be established. The pre-saccadic decrease in foveal sensitivity demonstrated previously as well as in our own data (*Figure 2B*) may boost the relative strength of fed-back signals by reducing the conspicuity of foveal feedforward input.

## What is the function of foveal prediction?

As stated above, previous investigations on foveal feedback required observers to make peripheral discrimination judgments. We, in contrast, did not ask observers to generate a perceptual judgment on the orientation of the saccade target. Instead, detecting the target was necessary to perform the oculomotor task. While the identification of local contrast changes would have sufficed to direct the eye movement, the orientation of the target enhanced foveal processing of congruent orientations. The automatic nature of foveal enhancement showcases that perceptual and oculomotor processing are tightly intertwined in active visual settings: planning an eye movement appears to prioritize the features of its target; commencing the processing of these features before the eye movement is executed may accelerate post-saccadic target identification and ultimately provide a head start for corrective gaze behavior (*Deubel et al., 1982*; *Ohl and Kliegl, 2016*; *Tian et al., 2013*). Since we routinely direct our gaze to relevant information rather than inspecting it peripherally, the foveation of peripheral input via an eye movement may not merely be another instance of foveal feedback, but the very reason for its existence.

## Conclusion

We suggest a crucial contribution of foveal processing to transsaccadic visual continuity which, up until now, has been overlooked. Feedback connections to foveal retinotopic cortex, so far characterized during passive fixation, gain a predictive nature during saccade preparation: they entail a retinotopic anticipation of soon-to-be foveated information – notably without the need for predictive spatial updating. As a behavioral consequence, the predictive foveal enhancement of target features demonstrated here may contribute to the continuous perception of eye movement targets and accelerate post-saccadic gaze correction.

## Materials and methods
### Sample

Seven human observers (five females, seven right-handed, four right-eye dominant, one author) aged 22–34 years (Mdn = 28.0) participated in Experiment 1 (see Supplements for sample size rationale). Nine observers, six of whom had completed Experiment 1, participated in Experiment 2 (seven females, all right-handed, seven right-eye dominant, no authors; 22–34 years, Mdn = 28.9). Seven observers, five of whom had completed Experiment 1, participated in Experiment 3 (five females, all right-handed, six right-eye dominant, one author; 23–36 years, Mdn = 29.0). Normal (n=4 in Experiment 1; n=5 in Experiment 2; n=5 in Experiment 3) or corrected-to normal (n=3 in Experiment 1; n=4 in Experiment 2; n=2 in Experiment 3) visual acuity was ensured at the beginning of the first session using a Snellen chart (*Hetherington, 1954*) embedded in a Polatest vision testing instrument (Zeiss, Oberkochen, Germany). Observers yielding scores of 20/25 or 20/20 were invited to proceed with the experiment. Ocular dominance was assessed using the Miles test (*Miles, 1930*). Since data collection was performed during the COVID-19 pandemic, our sample was composed of either lab members (n=6 in Experiment 1; n=8 in Experiment 2; n=7 in Experiment 3) or external participants recruited through word of mouth (n=1 in Experiment 1 and Experiment 2). Apart from the author, all observers were naïve as to the purpose of the study. Participants gave written informed consent before the experiments and were compensated with either accreditation of work hours, course credit or a payment of 8.50€/hr plus a bonus of 1€/session. The study complied with the Declaration of Helsinki in its latest version and was approved by the Ethics Committee of the Department of Psychology at Humboldt-Universität zu Berlin (reference number: 2018–09). The research question, experimental paradigm and data analyses were preregistered on the Open Science Framework (https://osf.io/xr2jk and https://osf.io/6s24m). Pre-processed data and experimental code are publicly available at https://osf.io/v9gsq/.

## Apparatus

### External setup

All experiments were conducted in a dark, sound-dampened booth. Stimuli were projected on a 200x113 cm screen (Celexon HomeCinema, Tharston, Norwich, UK) using a PROPixx DLP Projector (Vpixx Technologies, Saint-Bruno, QC, Canada) with a spatial resolution of 1920x1080 pixels and a refresh rate of 120 Hz (frame duration of 8.3ms). Observers faced the screen at a viewing distance of 180 cm while their heads were stabilized on a chin rest. The position of the dominant eye was recorded at a sampling rate of 1 kHz using a desk-mounted infrared eyetracker (EyeLink 1000 Plus; SR Research, Osgoode, Canada). Stimulus presentation was controlled by a DELL Precision T7810 Workstation (Debian GNU Linux 8) and implemented in Matlab 2016b (Mathworks, Natick, MA, USA) with the PsychToolbox (*Brainard, 1997*; *Kleiner et al., 2007*) and Eyelink toolbox (*Cornelissen et al., 2002*) extensions. Observers generated their responses on a standard QWERTY keyboard positioned centrally in front of them.

### Stimulus generation

To generate the background noise images, we applied a fast Fourier transform to uniform white noise, multiplied the noise spectrum with its inverse radial frequency, and transformed it back using an inverse fast Fourier transform (*Hanning and Deubel, 2022*; *Hanning et al., 2019*; *Hanning and Deubel, 2021*). For Experiment 1 and Experiment 3, 34 noise images (17 for the staircase and main experiment, respectively) were generated for each observer and session. While every noise image appeared in every trial, the order of presentation was randomized across trials. Within a trial, a certain noise image was never repeated. Assuming that nine noise images were presented between the onset of the noise stream and the onset of the saccade, and disregarding the order of presentation, a total of 24,310 image combinations (i.e., 17!/9!(17 − 9)!) were possible in the critical time window. Across all 833 images presented in the Experiment 1 (7 observers x7 sessions x17 images in the main experiment), the power spectral density (PSD) decreased by 5.07 dB/octave (*Figure 1B*). To allow for a larger variation of foveal noise content, we increased the number of images generated per session from 34 to 70 in Experiment 2. Only a random subset of all images appeared on a single trial. Assuming that nine noise images were presented between the onset of the noise stream and the onset of the saccade, and disregarding the order of presentation, a total of 70,607,460 image combinations (i.e., 35!/9!(35 − 9)!) were possible in the critical time window in Experiment 2. Across all 2205 images presented in the main experiment (9 observers x7 sessions x35 images), the power spectral density (PSD) decreased by 4.89 dB/octave.

To create the probe and target patches, we determined the background noise image that would be on screen at the desired probe presentation time and selected the pixel values in the relevant spatial region (3x3 dva) of that image. We subsequently filtered the orientation content of the noise window to –45° or 45° in the main experiment (*Figure 6*), and to –45°, 0°, or 45° in the staircase block (see Supplements). The width of the orientation filter representing the sharpness of orientation tuning was constant ($\alpha_{FiltWidth}$ = 20°). We adjusted the difficulty of the foveal detection task by varying the transparency of the probe against the noise background. To guarantee a smooth transition between the background and overlaid patches, we superimposed all patches with a raised 2D cosine mask (*Hanning and Deubel, 2021*; r=1.5 dva; σ=0.7 dva). While the probe appeared for the duration of a single noise image, the presentation of the saccade target spanned multiple images. Since the target constituted an orientation-filtered version of the respective background region, the appearance of the target changed dynamically in 50ms intervals.

## Procedure

Participants completed seven sessions within a mean span of 11 days in Experiment 1 and seven sessions within a mean span of 18 days in Experiment 2. Consecutive sessions were performed on separate days and lasted approximately 90 min. Each session started with a staircase block, followed by the main experiment. To familiarize observers with the task, the staircase was preceded by a slow-motion training block and an eye movement practice block in the first session. In all parts of the experiment, observers monitored the onset of a peripheral saccade target and, upon its detection, prepared an eye movement towards it. In 50% of trials, and at different time points before the saccade, a probe stimulus appeared. In Experiment 1, it appeared in the screen center, that is, observers' current,

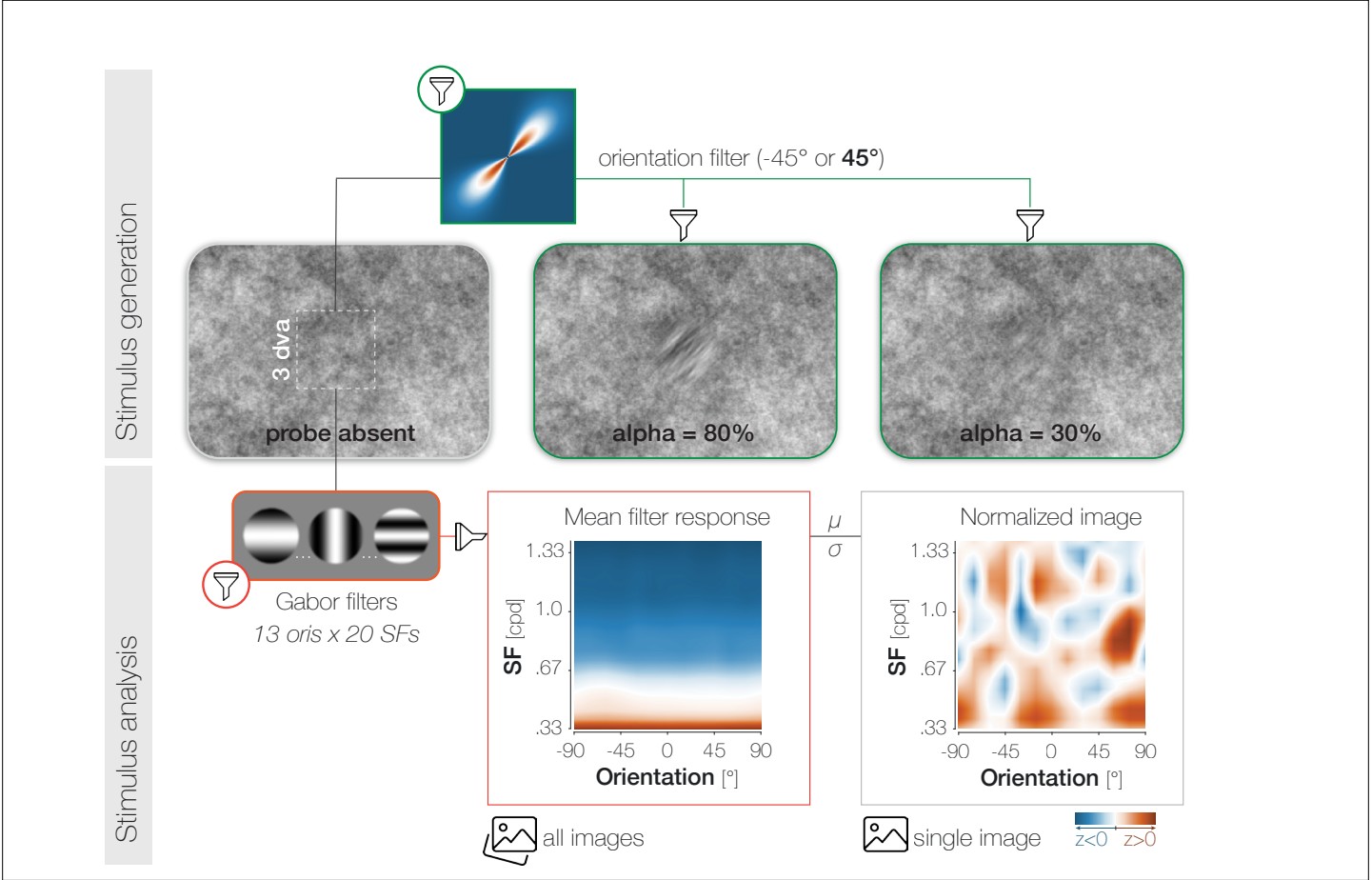

**Figure 6.** Stimulus generation (top) and analysis (bottom). To generate probe and target patches, we applied smooth orientation filters (green lines) to the relevant regions of the respective noise image. The 2D Fourier transform of a 45° filter is plotted for illustration. To adjust task difficulty for individual observers and different experimental purposes, we varied the opacity α (1-transparency) of the probe. High α-values increase visibility (see α=80%) while low α-values decrease visibility (see α=30%). Note that the calibration of the displaying device (in our case the projector) and the constant stimulus flicker in our experiment influence the visibility of the probe stimulus and should be considered when judging the displayed examples. To analyze the properties of the foveal noise window on probe absent trials, we computed its dot product with 260 Gabor filters with varying orientation (ori)*spatial frequency (SF) characteristics (yellow outlines). The resulting mean filter response across all images shows high energy for low-SF information (orange) and low energy for higher SFs (blue). To account for the asymmetry in power across SFs, we normalized every image by the mean (µ) and standard deviation (σ) of the set of images presented in a given experimental session for a given observer.

pre-saccadic center of gaze. In Experiment 2, it appeared anywhere between 4.5 dva to the left to 4.5 dva to the right of the screen center. Observers executed the eye movement and subsequently reported if the probe had been present or absent. In case of a 'present' response, they additionally reported its perceived orientation (2-AFC: left vs right).

Experiment 3 constituted a fixation control to Experiment 1 and involved a single session per participant. The main difference to Experiment 1 is that in Experiment 3, observers maintained fixation in the screen center throughout the trial.

## Session structure

### Task familiarization (session 1)

In the first session of each experiment, we familiarized observers with the task in a slow-motion and an eye movement training block (Supplements).

### Staircase procedure (all sessions)

Before the main experiment, we administered a staircase block to adjust the α-level, that is, the opacity (1–transparency) of the foveal probe against the background noise, to an optimal level. The

α-level was adjusted following a single-interval adjustment matrix protocol (SIAM; *Kaernbach, 1990*). A detailed description of the trial procedure and adjustment routine is provided in the Supplements.

## Experiment 1: Main experiment (all sessions)

At the beginning of each trial, a white fixation dot of diameter 0.3 dva was presented in the center of a static noise image covering the entire screen (*Figure 1A*). After stable fixation had been determined within a circle of 2.0 dva radius around the dot for at least 200ms, the first noise image remained visible for a random duration of 550–1050ms. Afterwards, the background noise started flickering at a temporal frequency of 20 Hz, corresponding to a presentation duration of 50ms for each image. Following a delay of 200ms after noise stream onset, the saccade target, i.e., an orientation-filtered patch with a diameter of 3 dva, appeared 10 dva to either the left or the right of fixation. Observers were instructed to move their eyes to the target as fast as possible. The target was oriented either –45° (corresponding to a leftwards tilt) or +45° (corresponding to a rightwards tilt) and appeared at an opacity of α=60% against the background noise. Since no additional saccade cue was presented, the detection of the target was necessary to perform the oculomotor task.

On 50% of trials, the probe, that is a second orientation-filtered patch of 3 dva diameter, appeared in the screen center. Crucially, the probe's orientation was either –45° or +45° and therefore either congruent or incongruent to the target. If present, the probe remained on screen for the duration of one noise image (50ms). To obtain sensitivity measures throughout the saccade preparation period while minimizing offline trial loss due to intra- and post-saccadic probe presentations, we defined three possible delays between target and probe onset. In the shortest delay condition, the probe was presented 50ms after target onset. The longest delay was set to an observer's median saccade latency in the preceding staircase block minus the duration of the probe (50ms). The intermediate delay fell right between the shortest and longest one. We slightly adjusted all delays, such that the on- and offset of the probe always coincided with the on- and offset of one background noise image in the stream. To obtain a baseline performance estimate outside the saccade preparation period, we added another onset condition in which the probe appeared 150ms *before* the target.

We investigated if foveating the saccade target briefly after saccade landing would influence pre-saccadic detection judgements. For this purpose, the experiment involved two trial types: on half of the trials (*Pre*), the saccade target was removed as soon as the recorded gaze position left a circle of 2 dva radius around the fixation dot. On the other half (*Pre-Post*), the saccade target remained visible for a short duration after gaze position had entered a circle of 3.25 dva radius around the center of the saccade target (see Supplements). Across all sessions and observers, the saccade target was foveated for Mdn = 22.6 ± 4.12ms after saccade offset (determined offline; *Figure 4C*).

Irrespective of probe and target timing, the background images flickered for a duration of 800ms. Afterwards, the last image remained on screen throughout the response period. Observers indicated if the probe in the screen center had been present or absent by pressing the up- or down-arrow key. After a 'present' response, they additionally reported the perceived orientation of the probe by pressing the left or right arrow key. We instructed observers to prioritize the presence/absence judgment over the orientation judgment. Specifically, they were encouraged to guess on the second response rather than responding 'absent' if they were unsure about the perceived orientation. The next trial was initiated after an intertrial interval of 500ms during which the last noise image remained on screen.

In each session, the main experiment was divided into six blocks of 107 trials each. Breaks were offered after every 54 trials. After the first block half, an observer's current HR and FAR along with the resulting d' score across all conditions was displayed on screen. In case of exceedingly good or poor performance, we adjusted the α-level of the probe and restarted the experiment (see Supplements).

## Experiment 2: Main experiment (all sessions)

To test if predictive enhancement is confined to the pre-saccadic center of gaze, we presented the probe on a horizontal axis of 9 dva length around the screen center (sessions 1–6). The experimental procedure remained unaltered with respect to Experiment 1, with the following exceptions:

First and foremost, the probe could appear in one out of 37 locations (randomly interleaved). Probe locations were spaced evenly on a horizontal axis from 4.51 dva to the left to 4.51 dva to the right of the screen center, in increments of 0.232 or 0.265 dva (i.e. 7 or 8 pixels). Second, we intended

to isolate pre-saccadic influences of saccade target features on foveal sensitivities and removed the target once gaze had crossed a boundary of 2.0 dva around the pre-saccadic fixation. Based on offline analyses, the target disappeared Mdn = 15 ± 4.30ms before saccade offset. Lastly, to obtain reliable spatial profiles within a feasible number of experimental sessions, we presented the probe at a single pre-saccadic time point, that is, 100–75ms before the eye movement. In this bin, congruency effects had been most pronounced in the *Pre* condition of Experiment 1 (see *Figure 4A*). Again, we relied on the median saccade latency in the preceding staircase block to time the onset of the probe in the main experiment. Based on offline analyses, the probe disappeared Mdn = 96.28 ± 18.13ms before the eye movement.

In session 7, we ensured that potential absence of enhancement for peripheral probes is a true consequence of probe location and cannot be attributed to baseline sensitivities. For this purpose, the probe was presented at one of two locations – 3 dva to the left or 3 dva to the right of the screen center (randomly interleaved) – and appeared at an α-value adjusted to approximate foveal performance. Observers were explicitly informed about the possible probe locations.

In sessions 1–6, the main experiment involved 8 blocks of 74 trials each. In session 7, the main experiment involved 9 blocks of 64 trials each. Breaks were offered between blocks. After the first block, we adjusted the α-level of the probe in case of exceedingly good or poor performance (see Supplements).

## Experiment 3: Main experiment

In contrast to previous experiments, observers maintained fixation within a circle of rad = 3.0 dva around the screen center throughout the trial. The target remained visible for 317.0ms, that is, the median target presentation duration in the *Pre* condition of Experiment 1. The probe could appear 50, 100, 150, 200, or 250ms after the target with equal probability and, like in the previous experiments, remained on screen for 50ms. Observers completed 10 blocks of 80 trials each.

In every experiment, we monitored observers' gaze behavior and aborted a trial if certain requirements were not met (see Supplements). Error-specific feedback messages were displayed after trial abortions. Aborted trials were appended at the end of a given block.

## Data analysis

### Eye movement pre-processing

The pre-processing of eye movement data and all subsequent analyses were implemented in Matlab 2018b (Mathworks, Natick, MA, USA). Saccades were detected offline using a velocity-based saccade detection algorithm (*Engbert and Mergenthaler, 2006*). We defined saccade onset as the time point at which the current eye velocity had exceeded the median eye velocity from all preceding samples by 5 SDs for at least 8ms. When recorded with pupil-based eye trackers, saccades often exhibit post-saccadic oscillations, which are assumed to reflect residual pupil movement rather than a true rotation of the eyeball (*Nyström et al., 2013*). We effectively excluded them from the saccadic profile by not merging detected saccadic events separated by one sample or more.

In offline analyses of Experiment 1, we excluded 8.12% of trials in which the recorded gaze behavior did not meet certain requirements (see Supplements). We excluded a further 4.62% of trials in which the foveal probe disappeared intra- or post-saccadically rather than before saccade onset. A total of 27,328 trials were carried on to further analyses. Trials were assigned to the *Pre* or *Pre-Post* condition based on an offline inspection of saccade and stimulus timing (*Figure 4C*): trials in which the target was supposed to disappear intra-saccadically but was still visible after the eye movement (mean n=9 trials per observer) were assigned to the *Pre-Post* condition. Trials in which the target was supposed to be visible after the eye movement but disappeared during saccadic flight were assigned to the *Pre* condition (mean n=42 per observer).

In Experiment 2, we excluded 11.25% of trials due to saccade characteristics and a further 2.29% due to intra- or post-saccadic probe presentations. We furthermore excluded trials in which the target had been visible after saccade landing (0.89% of trials). On all included trials, the target disappeared before saccade offset (sessions 1–6: Mdn = –15.00 ± 4.30ms; session 7: Mdn = –15.11 ± 5.33ms). A total of 31,674 trials were carried on to further analyses. The parameters of all included response saccades are provided in the Supplements. While saccade latencies and amplitudes showed

small-scale variations across probe locations in Experiment 2, these variations did not influence the spatial profile of enhancement.

In Experiment 3, we excluded 3.04% of trials in which observers had executed a microsaccade (amplitude ≤ 1.0 dva) or a saccade (amplitude >1.0 dva) before target offset. A total of 5,430 trials were carried on to further analyses.

## Analysis of noise content

We inspected the foveal noise content in probe absent trials separately for congruent FAs (observers reported perceiving the target orientation) and incongruent FAs (observers reported perceiving the non-target orientation). On each trial, we determined the noise images that had been visible for their full duration during the entire potential probe presentation period, that is, from the onset of the dynamic noise stream to the onset of the saccade (mean n=9.2). To isolate the impact of saccade preparation, we subsequently separated images that had preceded the onset of the saccade target (baseline; n=4) and images visible from the onset of the saccade target to saccade onset (saccade preparation; mean n=5.2; *Appendix 1—figure 3*).

To investigate the relation between response behavior and noise content at the potential probe location, we selected all pixels within a square of 3 dva side length around the screen center. We described the visual properties of each noise window along two dimensions: its spatial frequency (SF) and orientation (ori) content. To determine the energy of a certain SF*ori combination in the noise window, we created two Gabor filters with the corresponding properties [3 dva in diameter; one in sine and one in cosine phase (*Wyart et al., 2012*; *Li et al., 2016*; *Schweitzer and Rolfs, 2020*)]. We then convolved the pixel content in the noise window with both filters and averaged their responses. Note that Gabor filters with orientations from $-\pi/2$ to $\pi/2$ will yield slightly different responses than those with orientations from $-\pi/2+\pi$ to $\pi/2+\pi$ even though they effectively correspond to the same orientation (*Schweitzer and Rolfs, 2020*). We accounted for this by applying filters for orientations from $-\pi/2$ to $\pi/2$ as well as their counterparts from $-\pi/2+\pi$ to $\pi/2+\pi$, and averaging their responses (*Movellan, 2002*).

Using this method, we obtained filter responses for 260 SF*ori combinations per noise image (*Figure 6* in Materials and methods, 'Stimulus analysis'). SFs ranged from 0.33 to 1.39 cpd (in 20 equal increments). Orientations ranged from –90–90° (in 13 equal increments). To normalize the resulting energy maps, we z-transformed filter responses using the mean and standard deviation of filter responses from the set of images presented in a certain session. To obtain more fine-grained maps, we applied 2D linear interpolations by iteratively halving the interval between adjacent values 4 times in each dimension. To facilitate interpretability, we flipped the energy maps of trials in which the target was oriented to the left. In all analyses and plots,+45° thus corresponds to the target's orientation while –45° corresponds to the other potential probe orientation. Filter responses for all response types are provided at https://osf.io/v9gsq/.

## Experiment 1: Tests of statistical significance

We used bootstrapping to compare HRs and FARs between time points and congruency conditions. Within each observer, we determined the means in the to-be compared conditions and computed the difference between those means. Across observers, we drew 10.000 random samples from these differences (with replacement). Reported p-values correspond to the proportion of differences smaller than or equal to zero. We considered p-values ≤ 0.05 significant.

We used a two-step approach to identify SF*ori combinations that exhibited a significantly high or low energy in the foveal noise region when a congruent or incongruent FA was generated. First, we contrasted the energy of every SF*ori combination to zero (i.e. to the mean of the standard normal distribution for this specific combination), using two-sided one-sample t-tests. We then selected SF*ori combinations with a *t*-value >1.94 (corresponding to a significance level of p=0.05) and grouped neighboring above-threshold combinations into a cluster using the Matlab function bwconncomp (pixel connectivity = 4).

Two orientations were behaviorally relevant: –45° and +45°. To constrain our analyses, we therefore selected the two clusters with the highest summed *t*-value per response type and carried only those on to further tests. Indeed, all clusters identified in the first step were lateralized, i.e., exhibited centers of mass close to –45° or +45°. Since we had not corrected for multiple comparisons when identifying

the clusters, we verified their meaningfulness using a bootstrapping approach. Specifically, we tested whether filter energies within a cluster differed significantly from their SF-matched equivalents around the other relevant orientation. For this purpose, we determined the sum of filter responses within each cluster on an individual-observer level. To provide a test of lateralization, we flipped individual-observer maps in the orientation dimension – such that –45° now corresponded to +45° and vice versa – and summed the flipped filter responses within the non-flipped clusters. We contrasted the filter responses for original and flipped maps by drawing 10,000 bootstrapping samples from our set of observers (with replacement) and computing the mean difference in filter responses for each sample. A cluster was reported in the Results section if ≤5% of bootstrapped samples yielded a difference between original and flipped maps that was larger than the true difference. Reported p-values indicate this proportion. Three out of four clusters identified in the first step proved lateralized and are reported in the main text.

We would like to point out that, just like cluster-based permutation tests (*Maris and Oostenveld, 2007*; *Sassenhagen and Draschkow, 2019*), our approach does not establish the significance of single points within a cluster. While we can conclude that filter responses in a reported cluster differ significantly from filter responses in the same region of orientation-flipped maps, we cannot conclude that this only holds for a cluster with the exact dimensions reported. Identically to cluster-based permutations, statements on the precise extent of a cluster along the orientation and SF axes are thus descriptive rather than inferential in nature.

## Experiment 2: Function fitting

We determined congruent and incongruent HRs within a moving window including six adjacent locations (i.e. 1.46 dva; step size = 0.03 dva) as this ensured that each data point contained at least 30 trials on an individual-observer level. We subsequently fitted Gaussian functions with constant vertical offsets to the resulting spatial profiles. Before fitting, we realigned all probe locations to the mean fixation position recorded during the saccade preparation period (from target onset to saccade onset). We flipped probe locations in trials with leftwards saccades. In consequence, an x-axis value of zero indicates that the probe was presented in the center of gaze. Negative and positive x-values denote that the probe appeared in the opposite ('away') or same ('towards') hemifield of the saccade target, respectively. Details on the fitting routine are provided in the Supplements. The fitted profiles closely approximated observed HRs, with a mean absolute error of 0.5% for congruent and 1.1% for incongruent HRs. To allow for asymmetric profiles within each observer, we subsequently fitted two-term Gaussian functions to individual-observer data (*Appendix 1—figure 4*). Fitting two-term Gaussian functions did not alter the nature of findings: enhancement was highest in the center of gaze and significant within a similar spatial range. Since both approaches provided statistically equivalent fits (see Supplements), and since the Gaussian curve with vertical offset involves fewer yet more easily interpretable parameters, we present these fits in the Results.

Statistical comparisons were administered using bootstrapping (with replacement; n=10.000; see Experiment 1). Whenever we relied on a null effect to support a specific claim (e.g. the spatial specificity of enhancement), we supplemented p-values with Bayes Factors to gage evidence for the absence of an effect (*BF10*s, scale factor 0.707). Bayes Factor calculations rely on two-sided, one-sample t-tests. Only the test contrasting the peak parameters of the Gaussian profiles was directional (congruent >incongruent) since both potential mechanisms – spatially specific and global enhancement – predict this pattern.

## Experiment 3: Time course of enhancement

When a probe is presented at a certain time point during saccade preparation, two factors can influence the extent of enhancement at this time point: the duration for which the target had been visible before the probe appeared (target-probe interval) and the stage of saccade preparation at which the probe was presented. The time course during fixation, however, is influenced by the targe-probe interval only. To minimize the influence of saccade preparation when analyzing the target-locked saccadic time course (*Figure 5C*; left), we sorted every 'probe present' trial into two bins: one depending on the target-probe interval and one depending on when the probe had disappeared relative to saccade onset. To isolate the effect of the target-probe interval, we subsequently averaged across saccade preparation bins. In other words, every trial was sorted into one cell of a matrix

consisting of saccade preparation bins (rows) and target-probe intervals (columns). We then determined the average across all rows. Without this adjustment, the probe had disappeared at vastly different pre-saccadic time points across the four target-probe intervals ($m = -182, -129, -81$, and $-52$ms). After the adjustment, this range was reduced ($m = -142, -122, -92$, and $-69$ms). Note that a perfect balancing cannot be achieved since some time point combinations will naturally remain empty (e.g. the probe never appeared immediately after the target and at the same time immediately before the saccade). Conversely, to isolate the influence of saccade preparation irrespective of different target-probe intervals (*Figure 5C*; right), we determined the proportion of each target-probe interval in every pre-saccadic time bin in the saccade experiment and computed the inner product of this proportion and the measured enhancement per pre-saccadic bin (*Rolfs and Carrasco, 2012*).

## Acknowledgements

We thank Jude Mitchell and Shanna Coop for helpful discussions on foveal prediction, Lea Krätzig for assistance with data collection, and the members of the Active Perception and Cognition group for participating in our experiments in times of need. The article processing charge was funded by the Deutsche Forschungsgemeinschaft (DFG, German Research Foundation) – 491192747 and the Open Access Publication Fund of Humboldt-Universität zu Berlin.

## Additional information

### Funding

| Funder | Grant reference number | Author |
|---|---|---|
| Deutsche Forschungsgemeinschaft | RO3579/8-1 | Martin Rolfs |
| Deutsche Forschungsgemeinschaft | RO3579/9-1 | Martin Rolfs |
| Deutsche Forschungsgemeinschaft | RO3579/12-1 | Martin Rolfs |

The funders had no role in study design, data collection and interpretation, or the decision to submit the work for publication.

### Author contributions

Lisa M Kroell, Conceptualization, Data curation, Formal analysis, Investigation, Visualization, Methodology, Writing – original draft, Project administration; Martin Rolfs, Conceptualization, Resources, Supervision, Funding acquisition, Writing – review and editing

### Author ORCIDs

Lisa M Kroell ⓘ http://orcid.org/0000-0002-3508-5214
Martin Rolfs ⓘ http://orcid.org/0000-0002-8214-8556

### Ethics

Participants gave written informed consent before the experiments. All studies complied with the Declaration of Helsinki in its latest version and were approved by the Ethics Committee of the Department of Psychology at Humboldt-Universität zu Berlin (reference number: 2018-09).

### Decision letter and Author response

Decision letter https://doi.org/10.7554/eLife.78106.sa1
Author response https://doi.org/10.7554/eLife.78106.sa2

## Additional files

### Supplementary files

• Transparent reporting form

## Data availability

All data (psychophysical data, timing data, eye movement data, stimulus information) along with all experimental scripts are available on the Open Science Framework: https://osf.io/v9gsq/.

The following dataset was generated:

| Author(s) | Year | Dataset title | Dataset URL | Database and Identifier |
|---|---|---|---|---|
| Kroell L, Rolfs M | 2022 | Foveal sensitivity for saccade target-congruent and saccade target-incongruent orientation information during saccade preparation | https://osf.io/v9gsq/ | Open Science Framework, v9gsq |

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

## Appendix 1

### Supplemental Results

### HRs and FARs for individual observers in Experiment 1

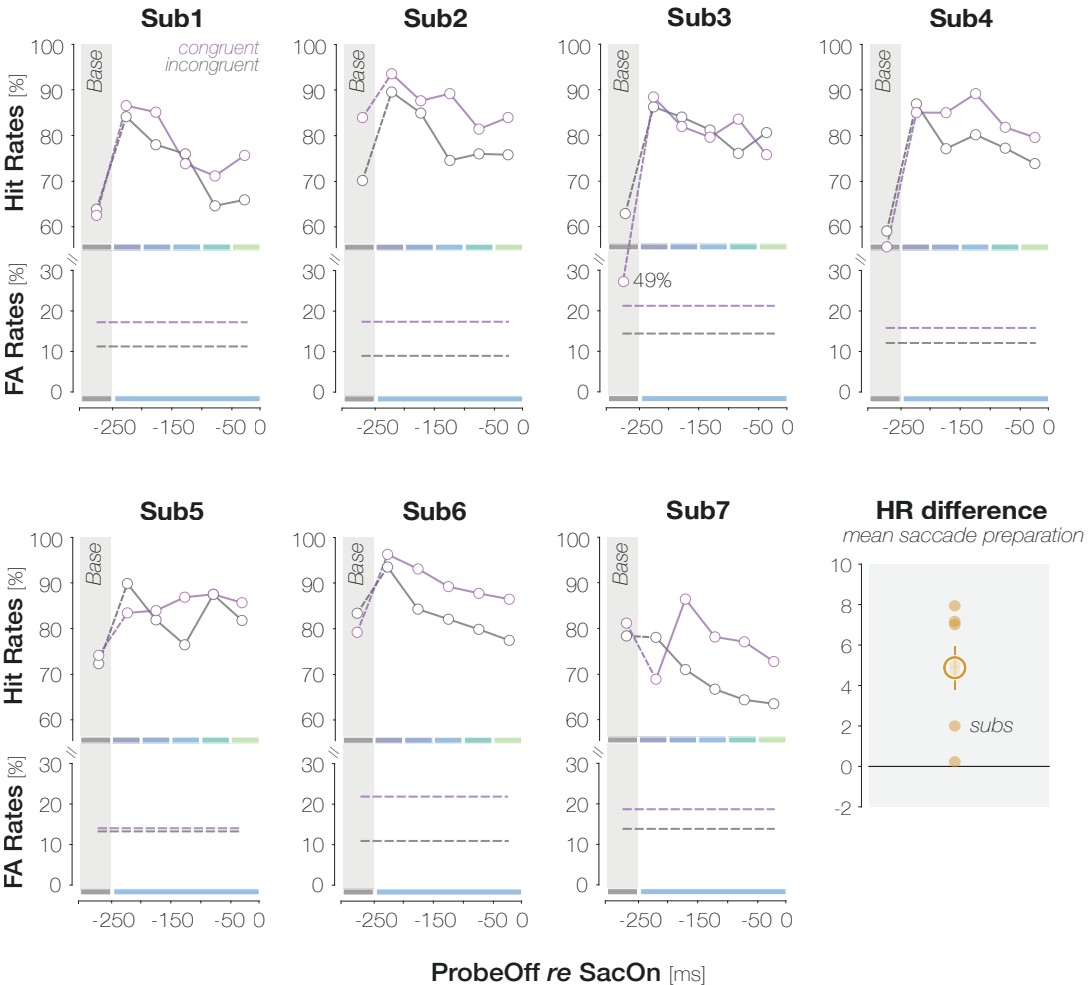

**Appendix 1—figure 1.** Congruency effects in HRs and FARs are fairly consistent across observers. Individual observer plots follow the conventions of *Figure 2B*. The last panel summarizes the congruency effect in HRs. Each dot corresponds to the mean enhancement (HR$_{cong-incong}$) across saccade preparation (all bins except baseline). The orange circle indicates the mean across observers. The error bar denotes ±1 SEM (n=7).

### HRs based on the accuracy of the orientation report

In our main analyses, we defined Hits as 'present' responses on probe present trials. We did not take the accuracy of the subsequent orientation report into account for two reasons: First, the difficulty of the foveal detection task had been adjusted to yield optimal performance levels for the presence/absence – not the orientation – judgment (see Materials and methods). Second, we instructed observers to prioritize the presence/absence over the orientation judgment and asked them to guess the orientation when they were uncertain rather than responding 'absent' altogether. Nonetheless, only defining 'present' responses after which the orientation of the probe had been reported correctly as Hits did not alter the nature of findings (*Appendix 1—figure 1*): While HRs in the baseline and earliest pre-saccadic time bin did not differ significantly (ps >.31), HRs for congruent probes significantly exceeded HRs for incongruent probes throughout the remaining bins (HR$_{cong-incong}$=6.7 ± 5.8%, 7.4 ± 7.0%, 7.3 ± 4.4%, 5.7 ± 5.7%, for the time bins [–200 –150[, [–150 –100[, [–100 –50[, [–50 0[ ms, respectively; all ps <.006). On average, observers misjudged the orientation

of the probe on 6.6% of Hit trials. On 58.6% of these misjudgment trials, observers falsely reported the target rather than the non-target orientation.

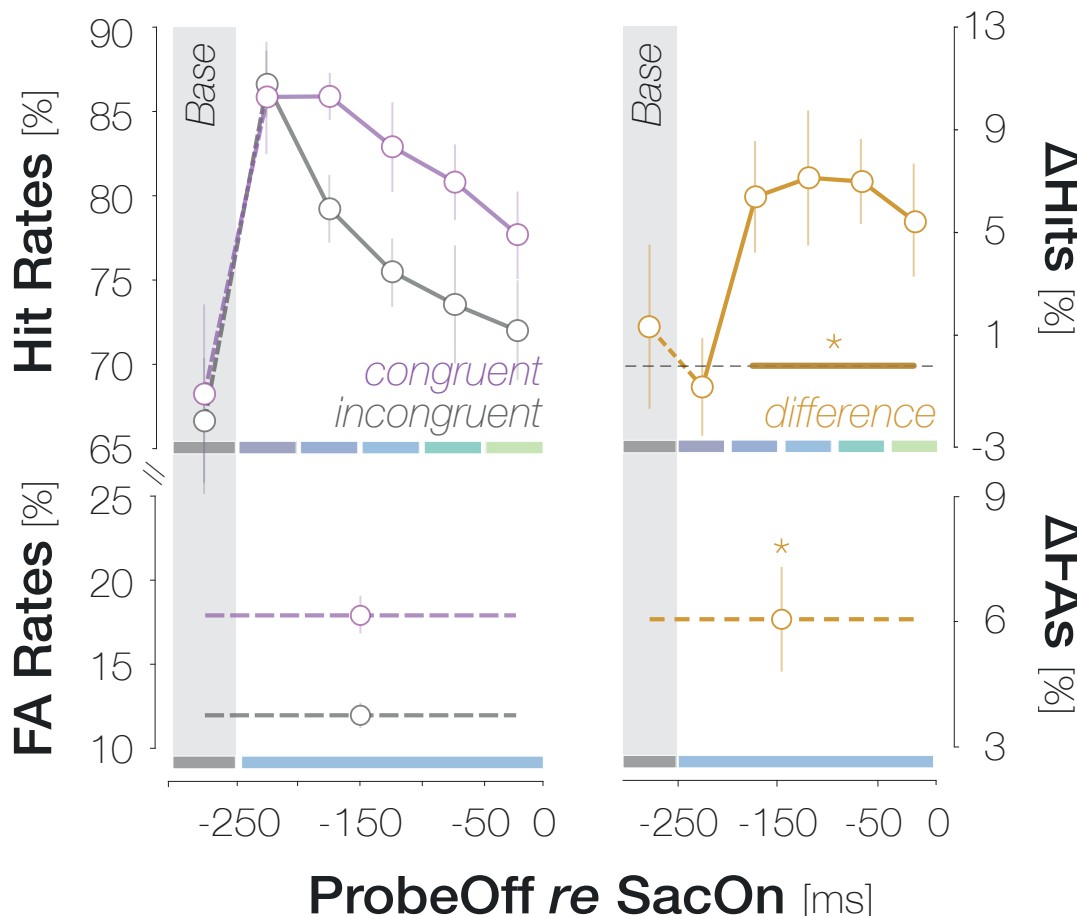

**Appendix 1—figure 2.** Taking the accuracy of the orientation report on Hit trials into account does not change the nature of findings. All conventions follow those of *Figure 2B*.

**Time-resolved noise image analysis**

# Time-resolved noise analysis

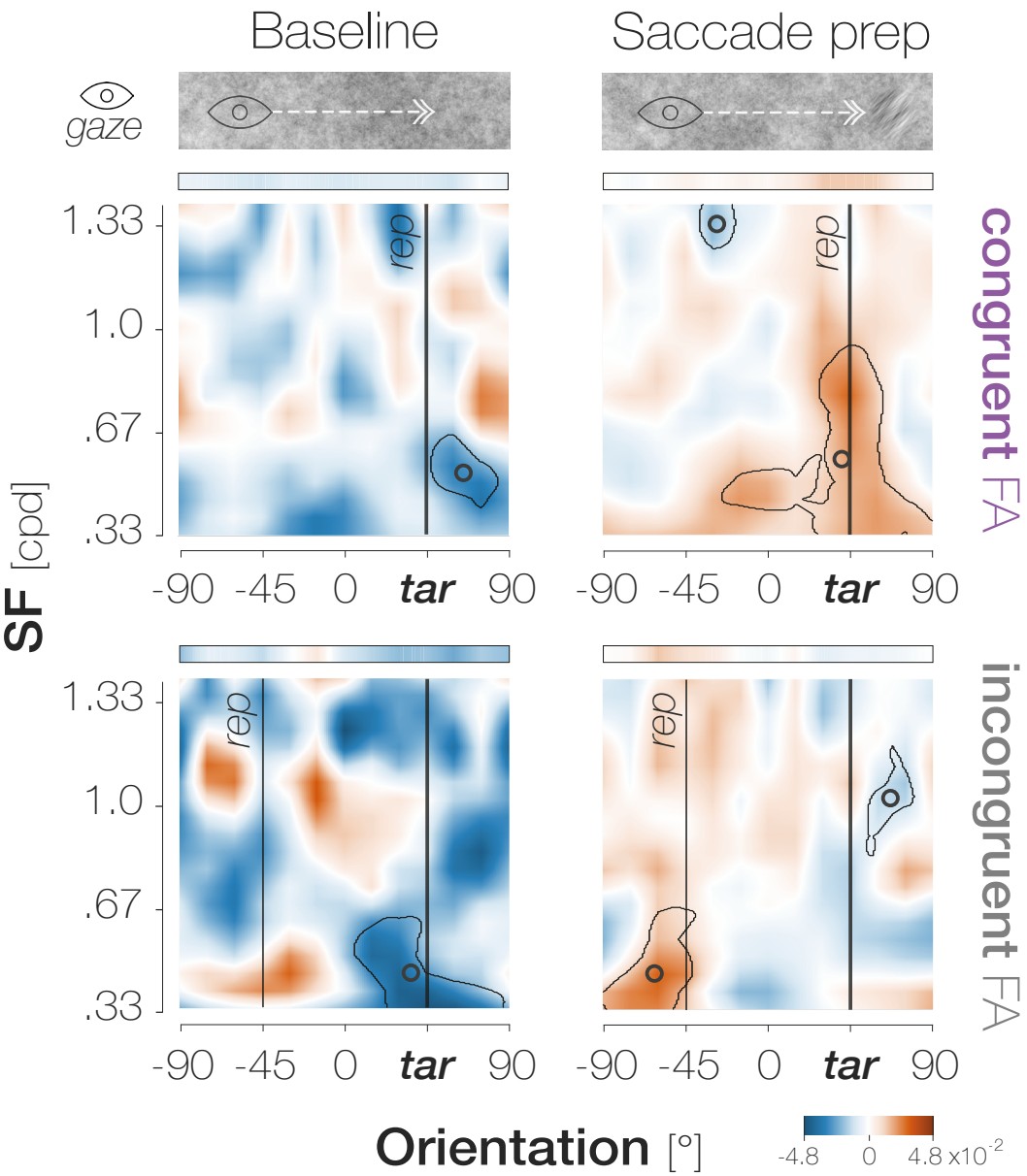

**Appendix 1—figure 3.** Higher-SF clusters around the target orientation manifest exclusively during saccade preparation. On every trial, we separated all presented noise images into those that had appeared before saccade target onset ('Baseline'; left column) and after saccade preparation had been initiated by the appearance of the target ('Saccade prep'; right column). For both congruent (top row) and incongruent (bottom row) FAs, clusters that include target-like orientations in a higher SF range are observable exclusively during saccade preparation (orange cluster indicating high energy for congruent FAs, blue cluster indicating low energy for incongruent FAs). See main text for further explanations. All conventions follow those of *Figure 2C*.

## Function fits to spatial HR profiles

We related HRs (y-axis) to probe locations (x-axis) using a Gaussian distribution with a constant offset from zero. All position values indicate horizontal coordinates.

HRs = yOffset + (yPeak - yOffset) * exp(-((prbLoc -xPeak) / Std) ^ 2) where *yOffset* refers to the constant vertical offset of the Gaussian curve, *yPeak* and *xPeak* refer to the vertical and horizontal coordinate of its peak, respectively, and *Std* refers to its standard deviation. Fitting was performed on an individual-observer level using a nonlinear least-squares fitting protocol (Matlab function 'lsqnonlin'; trust-region-reflective algorithm). We defined lower and upper bounds of 0 and 0.7 for *yOffset*, 0.3 and 1.0 for *yPeak*, 1 and 31 for *xPeak* and 0 and 31 for *Std*. To account for the reliability of each data point in the fitting process, we minimized the deviation between measured values and a weighted cost function. We obtained the weighted cost function by determining the deviations between observed HRs and HRs predicted with the current set of parameters at each iteration. We subsequently multiplied these deviations with a weight vector directly proportional to the number of trials contributing to a given data point. Across all observers and data points, assigned weights ranged from 0.62 to 1 for congruent HRs and from 0.63 to 1 for incongruent HRs. The minimum and maximum number of trials in an individual-observer data point amounted to 34 and 126, respectively. To obtain mean HR profiles across observers for plots, we determined a Gaussian curve with each observer's individual parameter estimates and averaged those curves.

While raw and mean fitted profiles exhibited a slightly asymmetric shape across observers, the fitted Gaussian curves were strictly symmetrical on an individual-observer level. To allow for an asymmetric shape of spatial profiles within each observer, we subsequently fitted two-term Gaussian functions to individual-observer data:

HRs = yPeak1 * exp(-((prbLoc - xPeak1) / Std1) ^ 2)+yPeak2 * exp(-((prbLoc - xPeak2) / Std2) ^ 2) where *yPeak1, xPeak1* and *Std1* denote the y-coordinate of the peak, the x-coordinate of the peak and the standard deviation of the first Gaussian curve while *yPeak2, xPeak2* and *Std2* denote the equivalent parameters of the second Gaussian curve. The resulting profiles are plotted in *Appendix 1—figure 4*. This fitting approach yielded a mean absolute error (MAE) of 0.7% for congruent and 1.0% for incongruent HRs and was therefore highly comparable to the more parsimonious Gaussian fit with vertical offset. A one-sample t-test on mean absolute errors revealed that the two fits were statistically indistinguishable, $t(8) = -1.64$, $P=0.139$.

In order to inspect the influence of saccade accuracy on the shape of the enhancement profile, we separated all trials into 'accurate' and 'inaccurate' saccades. For accurate saccades, the tails of the profiles were best captured by a mixture of Gaussians (MAE = 0.8%). The more parsimonious Gaussian functions with vertical offsets provided better fits to the spatial profiles of inaccurate saccades (MAE = 0.9% vs 1.0%). When probe locations were aligned to the remapped location rather than to the pre-saccadic center of gaze, Gaussian profiles could not provide adequate fits. These profiles were fitted with eight-order polynomials instead (MAE = 0.7%).

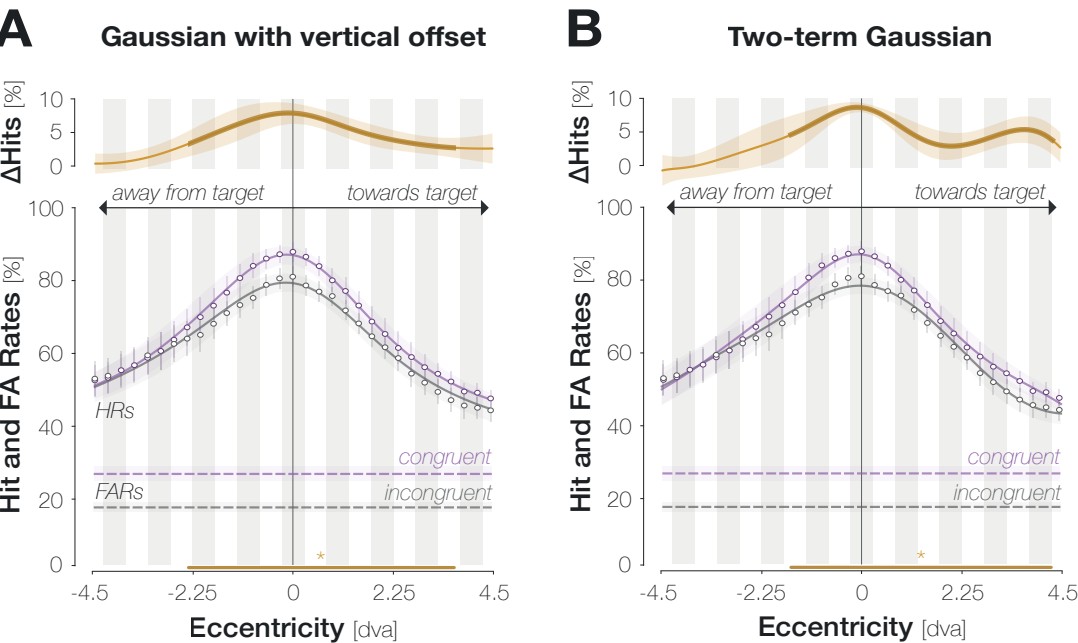

**Appendix 1—figure 4.** Gaussian functions with vertical offsets (**A**) and two-term Gaussian functions (**B**) yield comparable fits to spatial HR profiles. Conventions as in *Figure 2D*.

Note that the absolute width of the enhancement profile increases with the number of locations collapsed during analysis (*Appendix 1—figure 5*). Reducing the width of the boxcar window to two locations, i.e., the minimum number at which every observer contributed trials to every data point, yielded a slightly narrower enhancement profile that reached significance from –2.1 to 3.4 dva.

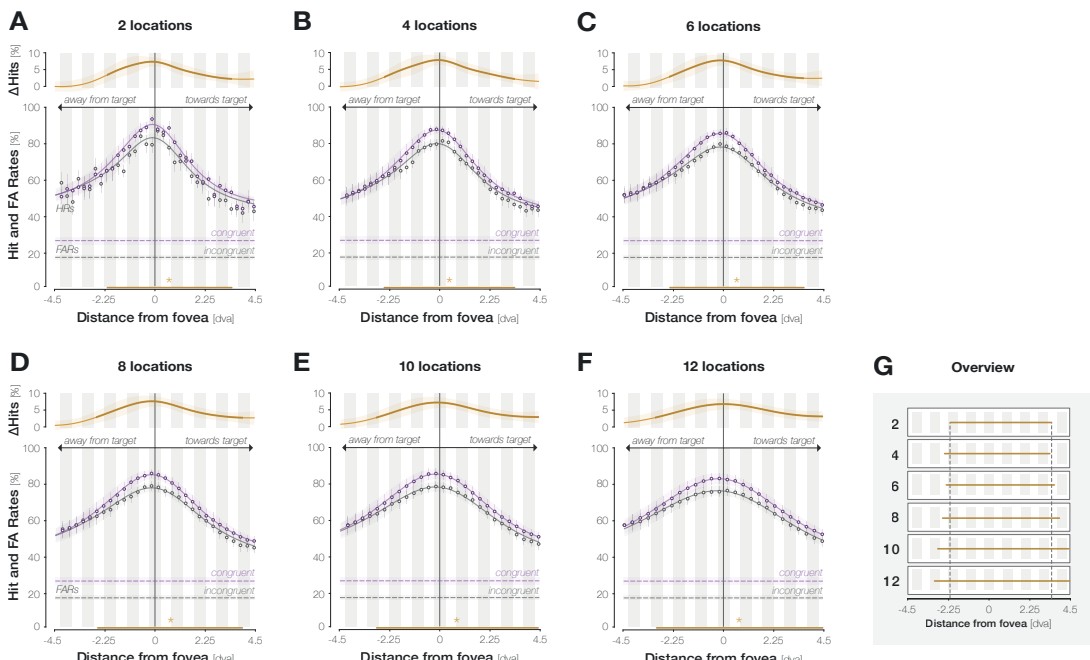

**Appendix 1—figure 5.** Spatial profiles for different widths of the moving boxcar window (i.e., different numbers of locations combined at each iteration). As the overview in (**G**) illustrates, the measured width of enhancement increases with the width of the moving window (row numbers). We used a window width of seven locations for the main analysis in *Figure 2D and a* width of nine locations when separating trials by saccade landing accuracy in *Figure 3*.

## Parameters of included saccades in Experiment 1

On average, observers executed hypometric saccades with a median amplitude of 9.30±.57 dva for leftwards and 9.74±.72 dva for rightwards saccades, and yielded a median saccade latency of 279.14±16.01ms (leftwards saccades: Mdn = 277.57 ± 23.19ms; rightwards saccades: Mdn = 279.21 ± 12.60ms). Saccade latencies were stable across stimulus and response conditions, suggesting that the presentation of the foveal probe did not alter eye movement preparation: we observed comparable latencies for probe present and probe absent trials (279.57ms vs 278.57ms, t(6) = 1.32, p=0.234), Hits, Misses, FAs and CRs (281.14 vs 275.29 vs 273.79 vs 280.71ms, $F(3,24)$ = 0.39, p=0.761, one-way ANOVA), Hit trials with target-congruent and target-incongruent foveal probes (284.29 vs 282.71ms, t(6) = 1.72, p=0.052), and FA trials with target-congruent and target-incongruent orientation reports (273.64 vs 273.00ms, t(6) = 0.34, p=0.748).

## Parameters of included saccades in Experiment 2

Observers executed hypometric saccades with a median amplitude of 9.37±1.09 dva for leftwards and 9.70±.96 dva for rightwards saccades. The median saccade latency was 271.50±26.40ms (leftwards saccades: Mdn = 273.00 ± 30.42ms; rightwards saccades: Mdn = 271.50 ± 22.15ms). Again, latencies were stable across stimulus and response conditions: we observed comparable latencies for probe present and probe absent trials (266.78 vs 269.33ms, t(8) = –1.90, p=0.094), Hits, Misses, FAs and CRs (266.67 vs 267.06.29 vs 268.79 vs 269.79ms, $F(3,32)$ = 0.03, p=0.994, one-way ANOVA), Hit trials with target-congruent and target-incongruent foveal probes (266.56 vs 267.11ms, t(8) = –0.351, p=0.735), and FA trials with target-congruent and target-incongruent orientation reports (268.56 vs 268.22ms, t(8) = 0.29, p=0.780).

Descriptively, saccade latencies and amplitudes showed small-scale variations across probe locations. To assess the significance of these variations, we performed two linear mixed-effects models in which we described the variance of amplitudes (*sacAmps*) or latencies (*sacLats*) with a fixed effect of probe location (*prbLoc*) and observer-specific, independent random effects for intercept and slope:

sacLats ~1 + prbLoc + (1 | sub) + (prbLoc | sub)

sacAmps ~1 + prbLoc + (1 | sub) + (prbLoc | sub)

For this purpose, we recoded probe positions such that the foveal location was assigned the largest value. Values assigned to the remaining locations decreased linearly with eccentricity. The fixed effect of probe location was non-significant for saccade latencies (t(277) = –1.35, p=0.177), with variations of less than 5ms across probe locations. Probe location did affect saccade amplitudes, t(277) = –2.89, p=0.004: amplitudes were shortest if the probe appeared in the fovea (min = 9.37 at *x*=0 dva) and increased with probe eccentricity. These variations in saccade amplitude, though systematic, ranged within 0.17 dva (~5 pixels).

Subsequently, we investigated whether saccade amplitudes, saccade latencies, or their respective interaction with probe location had a meaningful impact on congruency effects ($HR_{cong-incong}$). We added a random intercept of observer to account for inter-individual differences:

$HR_{cong-incong}$ ~sacAmp + sacLat +sacAmp:prbPos +sacLat:prbPos + (1 | sub)

None of the main effects or interactions reached significance (all ts <1.4, all ps >0.16), suggesting that the demonstrated spatial profile of enhancement is independent of the eye movement characteristics measured in our specific experimental design.

## Influence of saccadic precision on the width of the enhanced region

The width of the enhanced region may be related to saccadic precision on an individual-observer level. To investigate this, we estimated each observer's saccadic (im-)precision by computing bivariate kernel densities from their saccade landing coordinates. As we measured the horizontal extent of enhancement in our experiment, we defined the horizontal bandwidth as an estimate of saccadic imprecision. To estimate the size of the enhanced region for each observer, we created 10,000 bootstrapping samples for each observer's congruent and incongruent HRs. We then determined the difference between the bootstrapped congruent and incongruent HRs and considered enhancement at a certain location significant if ≤ 5% of these differences fell below zero. Finally, we defined the width of the enhancement profile as the maximum number of consecutive significant locations.

We observed a negative albeit non-significant correlation between the bandwidth of landing coordinates (i.e., saccadic imprecision) and the size of the enhanced window r=–.55, p=.129; *Appendix 1—figure 6*. In other words, we observed a non-significant tendency that the less precise

an observer's saccades, the narrower their estimated region of enhancement tended to be. We furthermore inspected the magnitude of enhancement per position within in the enhanced region. To do so, we computed the mean difference between congruent and incongruent HRs across all positions within the enhanced region. The sizes of the orange circles in *Appendix 1—figure 6* represent the resulting values (ranging from 3.2% to 10.9%). As saccadic precision decreased, the magnitude of enhancement per data point in the enhanced region tended to decrease as well. We therefore suggest that high saccadic precision is a sign of efficient oculomotor programming, which in turn allows peri-saccadic perceptual processes to operate more effectively.

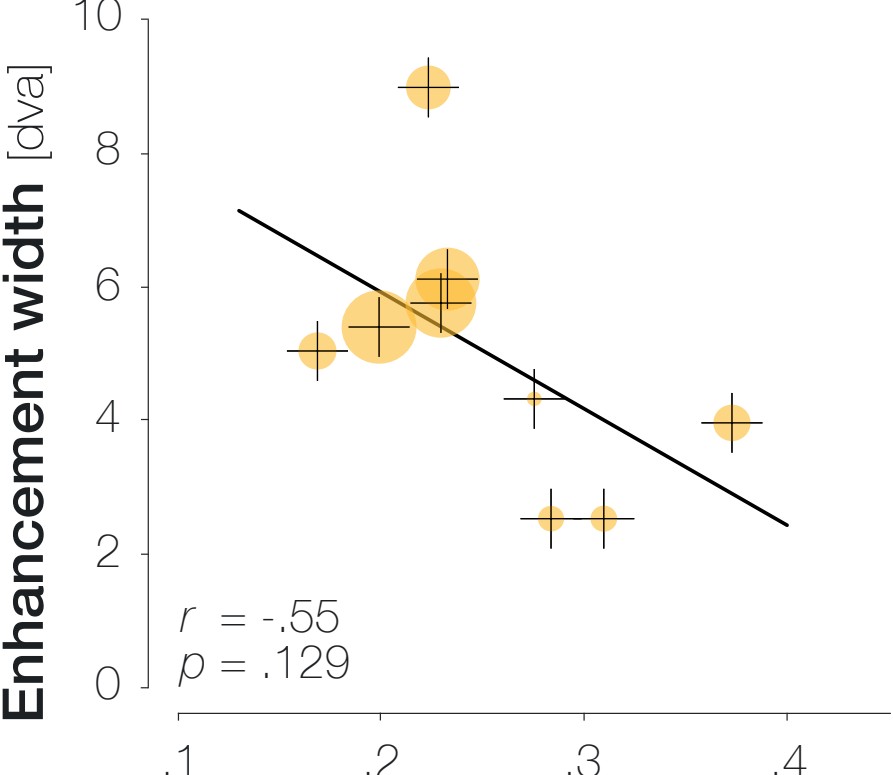

**Appendix 1—figure 6.** Relation between the imprecision of saccade endpoints (x-axis) and the width of the enhanced region (y-axis) in Experiment 2. Saccadic imprecision was defined as the bandwidth of the horizontal Gaussian kernel fitted to all saccade endpoints of a specific observer. X-axis values represent the kernel's full width at half maximum (FWHM) in dva. Enhancement width was defined as the maximum number of consecutive locations with significant enhancement. Orange disks correspond to individual observers. The size of each disk expresses the mean enhancement per significant data point (from 3.2% to 10.9%). The black line indicates a least-squares linear fit to the data.

## Spatial development of noise content

Our design innately provides spatial resolution by allowing us to relate response behavior to background noise properties at any desired display location. We made use of this possibility in an exploratory analysis and evaluated if – despite observers' explicit knowledge about the possible range of probe locations in Experiment 2 – FAs were primarily triggered by foveal orientation information. Specifically, we determined the energy around the reported orientation (target orientation ±22.5° for congruent FAs; non-target orientation ±22.5° for incongruent FAs) in the background noise along the axis of potential probe locations. Again, we collapsed all noise images that had appeared from the onset of the dynamic noise stream to saccade onset. Filter locations

matched the 37 experimentally defined probe locations and were combined into 31 moving windows just like probe locations were. Before the moving window analysis, energy values were normalized to the mean and standard deviation of all filter responses for that specific SF*ori combination at that specific location. A video displaying the spatial development of mean filter responses across spatial locations is provided as *Appendix 1—Video 1*.

For both congruent and incongruent FAs, the energy of the reported orientation was highest in or close to the center of gaze (congruent: 0 dva; incongruent: 0.30 dva; *Appendix 1—figure 7*). Energy values for congruent FAs significantly exceeded zero in a symmetrical range from –1.80–1.80 dva (all ps <.041; bootstrapping, 10,000 repetitions). Energy values for incongruent FAs significantly exceeded zero at two locations: 0 and 0.30 dva, ps <.048. The difference between the congruent and incongruent profile, however, did not reach significance at any location, all ps >.068. Note that the relation between a possible sensitization to target-congruent orientations and mean filter responses is complex and likely asymmetrical for congruent and incongruent information: That is, congruent FAs may readily be triggered by noise images with weak target-like orientation content, whereas incongruent FAs may require strong target-incongruent signals. While this pattern of results would be a consequence of sensitization, it would counteract any difference in normalized energy for congruent compared to incongruent responses.

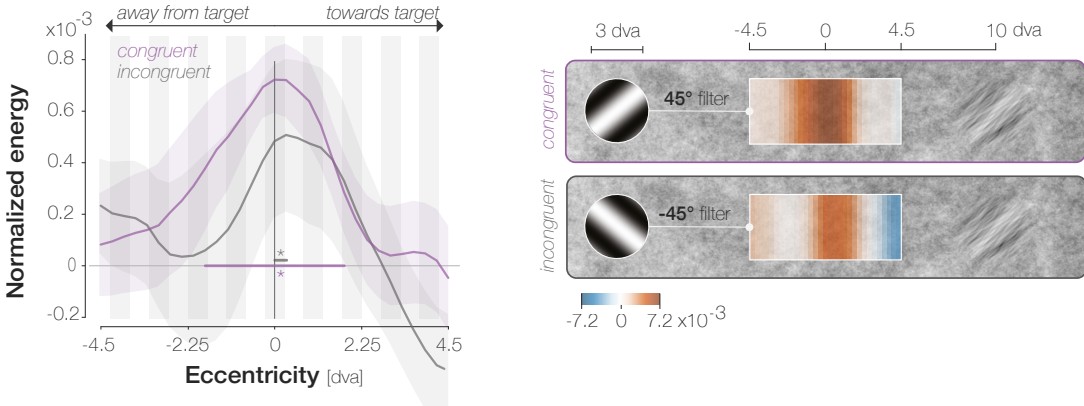

**Appendix 1—figure 7.** FAs are triggered by foveal orientation information. *Left*: Normalized energy around the reported orientation (+45° for congruent FAs, purple; –45° for incongruent FAs, gray) across spatial locations. Significance indicators highlight the spatial region in which the curve with corresponding color differs from zero. All further conventions follow those of *Figure 2D*. *Right*: Spatial profiles overlaid on the filtered region (drawn to scale with respect to the saccade target). Filters illustrate the relevant orientation in an example SF.

## Alignment of spatial profiles to the remapped target location

As explained in the main text, we aligned probe positions to the remapped target location on an individual-trial level. In addition to the main text, we inspected this alignment for all saccades irrespective of landing accuracy (*Appendix 1—figure 8*; left). Besides yielding flatter spatial profiles, this alignment reduced the spatial specificity of enhancement which now reached significance from –3.22 to 4.05 dva. These observations can be ascribed to the fact that the foveal location, in which both overall performance and enhancement is highest, now contributes to different x-axis values. For accurate saccades (*Appendix 1—figure 8*; right), the remapped target location is equal or close to the pre-saccadic center of gaze. Only then, the alignment was restored. We conclude that the enhancement of saccade target features is invariably centered on the pre-saccadic center of gaze – irrespective of the predictively remapped target location on an individual-trial level.

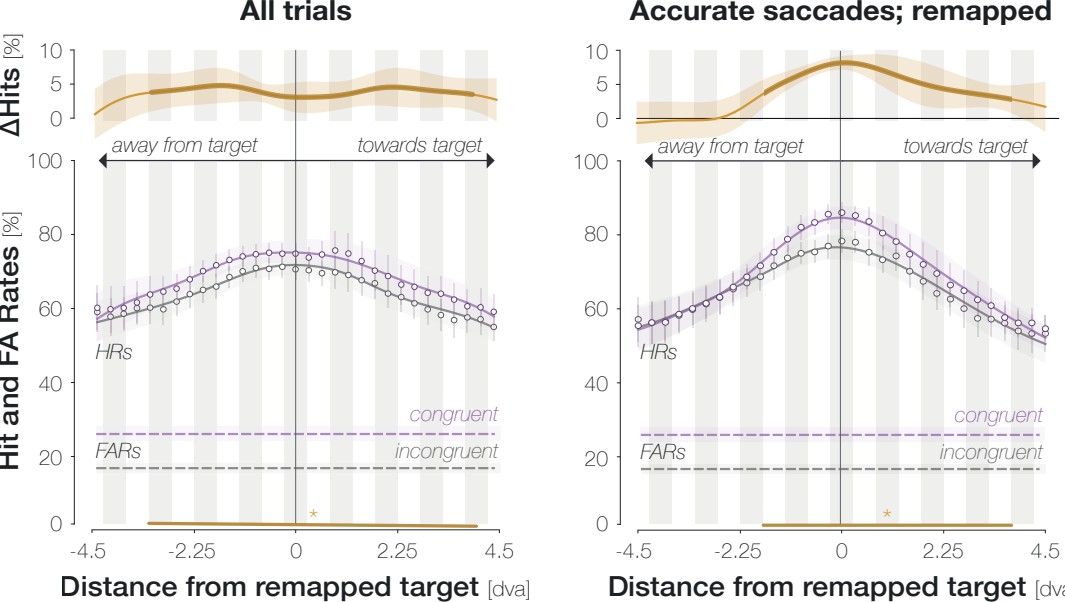

**Appendix 1—figure 8.** Spatial HR profiles aligned to the remapped location for all trials (left) and for the subset of accurate saccades that landed on the target (right). All conventions are as in *Figure 2D*.

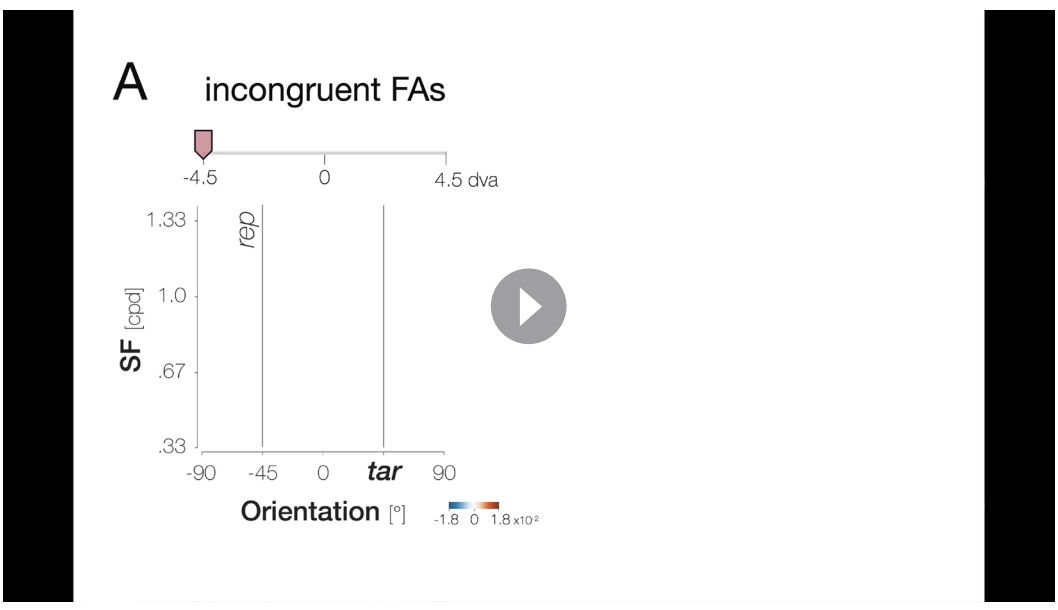

**Appendix 1—video 1.** FAs are primarily triggered by foveal orientation information. Filter responses of noise images underlying incongruent (A; the non-target orientation was reported) and congruent (B; the target orientation was reported) FAs at different spatial locations. The axis on top indicates the spatial range of filter locations (from –4.5–4.5 dva). The moving slider highlights the filter location corresponding to the currently displayed energy map. All further conventions are as in Figure 2C. For both FA types, the energy around the reported orientation is particularly high in and around the foveal region.
https://elifesciences.org/articles/78106/figures#video1

## Supplemental Methods

### Sample size rationale

Since we could not derive effect-size estimations from prior studies, we chose a sample size within the typical range of experiments investigating pre-saccadic attention shifts *White et al., 2015* for Experiment 1. We increased our sample size since we removed the target during the eye movement on all trials of Experiment 2. In Experiment 1, we had observed smaller congruency effects in this condition (*Pre*) than in the *Pre-Post* condition (across all pre-saccadic time bins: $HR_{cong-incong}$=2.52% vs 6.85%). Since Experiment 3 constituted a fixation control to Experiment 1, we collected seven observers to achieve an equal sample size.

### Task familiarization

In the first session of each experiment, we familiarized observers with the task by presenting a random subset of trials in a slowed down version and in the absence of oculomotor requirements. For this purpose, stimulus presentation times were increased by a factor of six. Participants generated verbal replies on the location of the saccade target, the presence or absence of the foveal probe, and its perceived orientation after a 'present' judgment. Once an observer was able to perform the task at the current speed, presentation times were gradually reduced until reliable task performance was achieved at normal speed. Observers subsequently performed eye movement practice trials until comfortable with the oculomotor aspect of the task.

### Staircase procedure

Before the main experiment in every session, we administered a staircase block to adjust the α-level, i.e., the opacity (1–transparency) of the foveal probe against the background noise, to an optimal level. The trial procedure was identical to the main experiment with the following exceptions: First, the α-level of the probe was adjusted adaptively in the staircase block but remained constant within each session of the main experiment. Second, to adjust the α-level in the absence of potential congruency effects, and to avoid rendering observers more familiar with the subset of incongruent trials before the main experiment, the saccade target was oriented vertically (0°) on all staircase trials. The probe stimulus was tilted 45° to the left or right, mirroring the main experiment. Third, while the probe was presented at one of four time points in the main experiment, it appeared 50ms after target onset on all staircase trials. We implemented this measure to avoid ceiling performance for incongruent probes in early stages of saccade preparation when we assumed foveal sensitivity to be highest. Lastly, the target was removed during the eye movement on all staircase trials. The α-level of the foveal probe was adjusted following a single-interval adjustment matrix protocol (SIAM; *Kaernbach, 1990*):

### Experiment 1

For each observer and session, we aimed to estimate the α-level at which a maximum reduced HR (HR –FAR) of 0.5 would be obtained. Initially, the probe was presented at an opacity of 50%. Possible α-levels ranged from 12% to 100%, in 2% increments. The α-level was adjusted after each trial based on the type of response generated: After Hits, it was reduced by 12%. After Misses and FAs, it was increased by 12% and 24%, respectively. No contrast adjustment was administered after Correct Rejections. The orientation report following a 'present' response did not affect the adjustment of α-levels. Initial step sizes were halved after the first and second reversal. Step sizes were reset if five consecutive Hits were generated at the same α-level. The staircase block terminated after 96 completed trials and took observers approximately 10 minutes to complete. The resulting opacity estimate was obtained by averaging α-values corresponding to the last six reversals. If fewer than six reversals had occurred, all available reversals were averaged to obtain the α-estimate.

### Experiment 2

No peripheral probes were presented in the staircase block – the contrast adjustment targeted an optimal performance level for the presence/absence judgment of foveal probes. Since we often had to reduce the estimated α-value to avoid ceiling performance in Experiment 1, we modified the classical SIAM protocol for our purposes: Initially, the foveal probe was presented at an α-level of 30%. Possible values ranged from 14% to 100%, in 2% increments. After each trial, the probe's α was adjusted depending on the type of response generated: After Hits, it was reduced by 18%. We did not increase α more after an FA than after a Miss, deviating from the classical SIAM protocol: after Misses and FAs, α was increased by 27%. We implemented this measure because FAs in our design

can indicate *increased* sensitivity to target-congruent orientation information in the foveal noise region. As the SIAM staircase assumes changes in FARs to result from changes in decision criteria rather than sensitivity to external signals, this property had likely inflated α-values in the staircase block of Experiment 1.

In the last session, we aimed to determine the α-level at which observers would yield the same incongruent HR for probes presented at 3 dva eccentricity as they did for a foveal probe in the preceding sessions. For this purpose, we randomly interleaved trials in which the probe was presented 3 dva to the left and 3 dva to the right of the screen center, rendering the location of the probe unpredictable on an individual-trial level. After Hits, the current α-value was reduced by 18%. After Misses and FAs, it was increased by 14.7%. No α-adjustment was administered after Correct Rejections.

In all sessions, initial step sizes were halved after the first and second reversal. Step sizes were reset if five sequential hits had been generated at the same α-level. The staircase terminated after 96 trials and took observers 10–15 minutes to complete. The resulting α-estimate was obtained by averaging α-values corresponding to the last eight reversals. If fewer than eight reversals had occurred, all reversals were averaged to obtain the α-estimate.

### Experiment 3
Just like in the main experiment, observers maintained fixation throughout the staircase block. The remaining parameters were identical to the improved adjustment protocol from Experiment 2.

### Foveal alpha adjustment
After the first block half of each experiment, an observer's current HR and FAR along with the resulting d' score across all conditions was displayed on screen. Performance measures were not labelled or otherwise interpretable for the observer. Prior to data collection, we had specified α-level adjustments in case of exceedingly good or poor performance (see preregistrations). In the course of data collection, we realized that, with increasing session number, observers tended to generate more FAs in the staircase block. This was likely the combined effect of repeated exposure to probes with high transparency and a training-induced increase in sensitivity to weak orientation content in the background noise. In general, we argue that FAs in our investigation reflect an enhancement of orientation information rather than purely unsystematic response behavior that would necessitate a decrease in task difficulty. To avoid a saturation of HRs in the main experiment, and to reduce the number of iterations needed to reach the targeted performance range, we therefore reduced the α-level by more than the preregistered step size (5%) at a time if considered appropriate. The experiment was restarted after every α-adjustment. In Experiment 1, the probe was presented at a median opacity of α=25.0 ± 2.9% (median per session: 31.4%, 24.0%, 25.0%, 25.0%, 25.0%, 25.0%, 22.0%). In sessions 1–6 of Experiment 2, the probe was presented at a median opacity of α=28.3 ± 7.6% (median per session: 33.0%, 28.9%, 29.6%, 27.8%, 25.2%, 28.2%). In the last session of Experiment 2 in which we targeted foveal-like performance levels for peripheral probes, the probe was presented at a median opacity of α=39.0 ± 11.1%. In Experiment 3 in which observers maintained fixation, the probe was presented at a median opacity of α=19.1 ± 4.1%.

### Gaze-contingent timing of target offset
In *Pre-Post* trials in Experiment 1, we intended to time the disappearance of the saccade target such that it would remain visible for 16–24ms (i.e., 2–3 refresh frames) after eye movement landing. Achieving this required an estimate of the time interval between the boundary cross and the true offset of the saccade determined after the testing session. As an initial estimation in the first session, we presented the target for 3 frames after the recorded gaze position first crossed a boundary of 3.25 dva around the target. After every session, we inspected the post-saccadic presentation duration in offline analyses and adjusted the number of target frames accordingly in the following session.

### Online trial abortion criteria
In Experiments 1 and 2, a trial was aborted if at any time before target onset, gaze position was recorded further than 2.0 dva away from the pre-saccadic fixation dot. After target onset, gaze position had to cross this 2.0-dva threshold within 400ms, in the direction of the target. Saccade landing had to be recorded within a circle of radius 3.0 dva around the center of the target. Note that target offset was time-locked to a slightly larger boundary (3.25 dva) which had achieved most

accurate timing during piloting. To control post-saccadic foveal input, observers' gaze position had to remain within a circle of radius 5.5 dva around the target until the first keyboard response had been generated. In Experiment 3, a trial was aborted if gaze position was recorded outside a circle of 3 dva radius around the screen center.

## Trial exclusion criteria in offline gaze analyses

In Experiments 1 and 2, we removed trials in offline analyses if no response saccade had been generated in the critical time window between 150ms before and 550ms after cue onset, if a saccadic event was detected before the response saccade and/or if gaze position samples were missing anytime before saccade onset. Moreover, trials involving anticipatory response saccades, i.e., saccades with a latency below 80ms, were excluded. To ensure that post-saccadic foveal input was stable on the retina for a certain period of time after saccade landing, we excluded trials in which a second saccade had occurred between 25 and 100ms after response saccade offset ('crit' in *Figure 4C*). Since post-saccadic oscillations were often registered as a second saccadic event, we introduced a short time window of 25ms after response saccade offset, during which saccadic activity did not lead to trial exclusions. In Experiment 3, we excluded all trials in which observers had generated a (micro-)saccade and/or if gaze position samples were missing anytime before target offset.

