## [Editor Report]

In a methodologically sophisticated study of pre-saccadic processing at the fovea, Kroell and Rolfs provide compelling evidence that saccade preparation causes feature-specific pre-saccadic visual enhancement restricted largely to the center of gaze. The authors were able to differentiate this effect from pre-saccadic enhancement during passive fixations and to rule out criterion shifts as a mechanistic explanation. The fundamental implication of these findings will be of interest to both vision scientists and modelers. They parametrize a potential mechanism for visual continuity across saccades, with foveal processing identified as a key, contributing component.

---

## [Decision Letter]

**Decision letter after peer review:**

Thank you for submitting your article "Foveal vision anticipates defining features of eye movement targets" for consideration by *eLife*. Your article has been reviewed by 3 peer reviewers, one of whom is a member of our Board of Reviewing Editors, and the evaluation has been overseen by Chris Baker as the Senior Editor. The reviewers have opted to remain anonymous.

Essential revisions:

Included here is a brief evaluation summary and a list of revisions the reviewers and review editor deem essential for the authors to address. The public summaries and full, individual reviewers' recommendations for the authors are also appended below. The authors are advised to address the public summaries briefly, and the individual recommendations in a detailed, point-by-point manner.

As you will be able to read below, all reviewers appreciated the study and manuscript describing it as a potentially very valuable contribution to the field of vision science. The writing was clear, the figures elegant and importantly, the study design and analyses were deemed rigorous, generally appropriate and elegant. The insights and data presented – especially if strengthened as detailed below – should be highly interesting to those studying active vision, as well as those studying low-level and high-level visual perception. However, as you will also see, reviewers raised some significant concerns with regard to the claims and data interpretation. Perhaps these issues cannot be addressed without collecting additional data, or at least additional analyses. However, given the interest in the research question and the value of the dataset, it was agreed that we would like to give the authors a chance at rebuttal.

In addition to the recommendations listed below in individual reviews, the points that need to be addressed can be summarized as follows:

1. More careful use of terminology related to the foveal region analyzed: please specify throughout whether you are referring to the foveola/central fovea, or the fovea. This is crucial to reduce confusion, and given implications in terms of different circuitry and neurophysiology between these regions, will impact the interpretation of the results.

2. The paper's main limitation is that it appears to entertain only one hypothesis (that saccade preparation enhances sensory processing) to explain the findings. But in fact, there are several alternative hypotheses, including the possibility of a criterion shift, purely sensory enhancements related to fixation duration, enhancements related to covert attention (and not saccade preparation), among others. These are not considered in the manuscript, and they should be.

For instance, the addition of a no-saccade experimental condition could help assess if spatial selectivity of enhancement remains the same as during the saccade task already reported. We realize that this would require collecting additional data on subjects, but at some level, this comparison may be what is needed to critically test the authors' current/sole hypothesis.

In addition, it would be good to see an assessment/discussion of whether a criterion shift may explain the present results. This could be tested by performing a quantitative comparison between two 2-stage models: one with sensory gain and one with criterion shifts at the decision stage.

In sum, at a minimum, the authors should outline and discuss other possible hypotheses, and this may ultimately cause them to tone down their main claims. Alternatively, they should provide data or modeling evidence (i.e., a convincing rationale) as to why these other hypotheses should be ruled out.

*Reviewer #2 (Recommendations for the authors):*

I thought the paper was clearly written and the figures were gorgeous. My concerns are outlined in the public review, but I really do think more needs to be done to address the issue of "enhancement". I have a few ideas for how this could be achieved.

The best idea I have is to use the external noise and project each frame on either the "optimal template" or on a "subject-specific template" to get a single 2D decision variable. I use the term "optimal" loosely, but the ideal observer would have a template that is matched to the target (orientation / SF energy). Now, there are two possible targets and a present/absent judgement, so it's a 2D task. But by projecting on a single template (or set of two templates) you either end up with a 1D variable or a 2D variable and then you can condition on that to calculate something like a d-prime. It involves a few assumptions, but I think they're minimal and reasonable. You want to show that based on this signal, the subjects are making choices more accurately. This is like looking at a spatiotemporal psychophysical kernel using the full reverse correlation, but collapsing across space (orientation and SF) using a template.

Another idea is to use a computational model of the choices to show that this must be a gain change. You'd have to build the same type of observer model (linear template -> decision variable -> criterion), but then you could play with how gain on the templates change vs how criterion shifts change. I think this level of additional analysis is really necessary here because enough about this task is new that it really needs these alternative models quantitatively examined. In the best case, I would lay out the different hypotheses specifically and quantitatively compare them.

Similarly, the spatial tuning of enhancement needs something to show that it's not just a static gain on an already tuned mechanism. That could easily be done with a few subjects doing the task without a saccade. Or more extensive modeling work could tease apart the shape of effects under different hypotheses.

On a smaller note, the language about the "fovea" can be confusing. In the anatomical literature (which is referenced in the 0.01% to 8% over-representation number in the intro), the "fovea" refers to the entire pit in the retina, which subtends more than 5 degrees of visual angle. The "foveola" is the rod-free zone that takes up only 0.01%. I realize that neurophysiologists are often sloppy about this distinction and it detracts from a simple "foveal" narrative, but it kind of continues the sloppiness in the field. I wonder if being precise about what exactly you mean by "fovea" would be that much of a hindrance to the writing? It does matter for thinking about the circuitry because the one-to-one projections in the fovea extend well beyond the central one degree. So what really is special about the foveola (lots of things, but it's not spelled out here)?

One paper of potential relevance for the discussion is "Transsaccadic integration of visual information is predictive, attention-based, and spatially precise" by Wilmott and Michel.

https://jov.arvojournals.org/article.aspx?articleid=2776566

[Editors' note: further revisions were suggested prior to acceptance, as described below.]

Thank you for resubmitting your work entitled "Foveal vision anticipates defining features of eye movement targets" for further consideration by *eLife*. Your revised article has been evaluated by Chris Baker (Senior Editor) and a Reviewing Editor.

The manuscript has been improved but there are some remaining issues that need to be addressed, as outlined below:

All three reviewers appreciated the manner in which you were able to address the great majority of their comments, and the additional experiments performed, which were deemed to add significant value to the study.

Reviewers were reasonably convinced by your arguments that criterion shifts are unlikely to explain the phenomena at hand. However, they requested that you modify your Discussion to include consideration of alternative mechanisms. Specifically, please respond to the following concerns, which revolve around the notion that dismissing criterion shifts as explaining the obtained results because of spatial/temporal specificity might be too simplistic an interpretation:

1. Please discuss whether the particular task design used in the present study could cause spatial distributions of hit rates and false alarm rates that might be incorrectly interpreted as enhancement, as suggested in the work of Sridharan et al., JNeurosci (2017).

2. Additionally, based on the extensive literature on temporal changes in criterion during decision-making (i.e., "collapsing bounds" or "urgency signals"), please discuss if there could there be urgency signals during saccade preparation that lead up to a decision and then saccade generation? Is it possible that the reason for an increase, then a decrease in hit rates pre-saccadically is a temporal change in criterion until saccade generation is hit (i.e., a bound is crossed)?

---

## [Author Response]

Reviewer #2 (Recommendations for the authors):I thought the paper was clearly written and the figures were gorgeous. My concerns are outlined in the public review, but I really do think more needs to be done to address the issue of "enhancement". I have a few ideas for how this could be achieved.The best idea I have is to use the external noise and project each frame on either the "optimal template" or on a "subject-specific template" to get a single 2D decision variable. I use the term "optimal" loosely, but the ideal observer would have a template that is matched to the target (orientation / SF energy). Now, there are two possible targets and a present/absent judgement, so it's a 2D task. But by projecting on a single template (or set of two templates) you either end up with a 1D variable or a 2D variable and then you can condition on that to calculate something like a d-prime. It involves a few assumptions, but I think they're minimal and reasonable. You want to show that based on this signal, the subjects are making choices more accurately. This is like looking at a spatiotemporal psychophysical kernel using the full reverse correlation, but collapsing across space (orientation and SF) using a template.Another idea is to use a computational model of the choices to show that this must be a gain change. You'd have to build the same type of observer model (linear template -> decision variable -> criterion), but then you could play with how gain on the templates change vs how criterion shifts change. I think this level of additional analysis is really necessary here because enough about this task is new that it really needs these alternative models quantitatively examined. In the best case, I would lay out the different hypotheses specifically and quantitatively compare them.

We thank the reviewer for their very insightful and detailed suggestions. Unfortunately, our stimulus material is not cut out for noise analyses at this level of complexity. In Experiment 1, in which we presented exclusively foveal probes, and which would therefore be most diagnostic, we presented the same 17 noise images on every trial of a given session and merely varied their order (such that a different subset would be presented during saccade preparation across trials). The differences in noise properties between trials in general and between response types in particular become apparent when averaging over a large number of trials and time points. Overall, however, they are small. To tease out the influence of a target template on potential criterion shifts, we would need to make several assumptions: since we presented our probe at one of three pre-saccadic time points which were determined gaze-contingently, it should have been almost impossible for the observer to predictively prioritize a certain saccade preparation window. While desirable in general, this prevents us from narrowing down our analyses to a critical subset of informative noise images. We could use the demonstrated time course of enhancement to constrain the set of noise images. However, the enhanced window covers medium and late stages of saccade preparation and varies depending on whether the target had been removed intra-saccadically, which would necessitate further trial separations. Moreover, we have since demonstrated that the spatial frequency spectrum of the foveally enhanced signal depends on the visual resolution of the peripheral saccade target (VSS abstract to be available in JoV). Since target resolution varies dynamically during saccade preparation (Kroell and Rolfs, 2021), we may need to adapt the template dynamically throughout a trial and make assumptions on the information transmission time from the target to the foveal location. In short, the small variance in stimulus properties across trials would require us to make a set of additional assumptions to be able to tease out meaningful differences and make conclusions on a variation of criterion or sensitivity. Even if we were able to conclusively distinguish between those hypotheses based on noise image properties only, the number and flexibility of assumptions required to reach this conclusion would make us doubt its validity. As mentioned in the public replies, we had always planned to exclude a criterion shift experimentally by demonstrating specificity in HRs.

Similarly, the spatial tuning of enhancement needs something to show that it's not just a static gain on an already tuned mechanism. That could easily be done with a few subjects doing the task without a saccade. Or more extensive modeling work could tease apart the shape of effects under different hypotheses.

We fully agree. However, we do not consider a fixation condition diagnostic to resolve this question since, as of now, correlates of foveal feedback have exclusively been observed during fixation. In those studies, it was suggested that the effect, i.e., a foveal representation of peripheral stimuli, reflects the automatic preparation of an eye movement that was simply not executed (Fan et al., 2016; Chambers et al., 2013; Yu and Shim, 2016). To address another reviewer’s comment, we collected additional data in a fixation experiment. The probe stimulus could exclusively appear in the screen center (as in Experiment 1) and observers maintained fixation throughout the trial. While pre-saccadic congruency effects were significantly more pronounced and developed faster, congruency effects did emerge during fixation when the probe appeared 200 ms after the target. If pre-saccadic processes indeed spill over to fixation tasks to some extent and trigger relevant neural mechanisms even when no saccade is executed, we could expect a similar feedback-induced spatial profile during fixation. Since this matches the reviewer’s prediction if the pre-saccadic profiles resulted from inhomogeneous feedforward processing, we do not consider a fixation condition suitable to distinguish between both hypotheses.

To test whether the tuning of enhancement is effectively a consequence of declining visual performance in the parafovea/periphery, we instead raised parafoveal performance to a foveal level by adaptively increasing the opacity of the probe: while leaving all remaining experimental parameters unchanged, we presented the probe in one of two parafoveal locations, i.e., 3 dva to the left or right of the screen center. Observers were explicitly informed about the placement of the probe. We administered a staircase procedure to determine the probe opacity at which performance for *parafoveal* target-incongruent probes would be just as high as *foveal* performance had been in the preceding sessions. While the foveal probe was presented at a median opacity of 28.3±7.6%, a parafoveal opacity of 39.0±11.1% was required to achieve the same performance level. As a result, the gray dot at 0 dva in Author response image 1 represents the incongruent HR in the center of gaze and ranges at 80% on the y-axis. The gray dots at ±3 dva represent incongruent parafoveal HRs and also range at ~80% on the y-axis. Using the reviewer’s terminology, we effectively removed the influence of acuity- (or contrast-sensitivity-) dependent spatial tuning. If the spatial profiles had indeed been the result of “global feature gain on top of already spatially tuned processing”, this manipulation should render parafoveal feature gain just as detectable as foveal feature gain. Instead, congruent and incongruent parafoveal HRs were statistically indistinguishable (away from the saccade target: *p* = .127, *BF10* = 0.531; towards the saccade target: p = .336, *BF10* = 0.352), inconsistent with the idea of a spatially global feature gain.

**Author response image 1. sa2fig1:** 

We had included these data in our initial submission. They were collected in the same observers that contributed the spatial profiles (Experiment 2). The data points at 0 dva in the reduced figure above correspond to the foveal probe location in Figure 2D. The data points at ±3 dva had been plotted and discussed in our initial submission, yet only very briefly. Based on this and another reviewer’s comment, we realize that we should have explained this condition more extensively in the main text rather than in the Methods and have added a dedicated paragraph to the Results section.

On a smaller note, the language about the "fovea" can be confusing. In the anatomical literature (which is referenced in the 0.01% to 8% over-representation number in the intro), the "fovea" refers to the entire pit in the retina, which subtends more than 5 degrees of visual angle. The "foveola" is the rod-free zone that takes up only 0.01%. I realize that neurophysiologists are often sloppy about this distinction and it detracts from a simple "foveal" narrative, but it kind of continues the sloppiness in the field. I wonder if being precise about what exactly you mean by "fovea" would be that much of a hindrance to the writing? It does matter for thinking about the circuitry because the one-to-one projections in the fovea extend well beyond the central one degree. So what really is special about the foveola (lots of things, but it's not spelled out here)?

We appreciate and sympathize with this point regarding the precision of the language in the manuscript (and the field in general). Being precise is not a hindrance to the description of our own results. Yet, it had proven difficult to achieve this precision when reviewing existing literature. We thank the reviewer for their correction and have adapted the anatomical reference in the Introduction. In general, our probe stimulus had a diameter of 3.0 dva and therefore extended past the foveola. This likely influenced the width of the spatial profiles, as we now clarify in the Results and Discussion. Irrespective of its width, enhancement peaked at 0.15 dva, suggesting highly localized feedback. Nonetheless, since the probe covered both the foveola and parts of the surrounding foveal region, we would not like to state that our findings exclusively pertain to the foveola. For these reasons, we would like to maintain the expression “fovea” or “center of gaze”. Nonetheless, we fully agree with the reviewer that we should make this reasoning explicit to avoid future confusion. We now do so in the Introduction:

“The foveola covers the central 1.3 degrees of visual angle (dva) (Hendrickson, 2005). The fovea and parafovea cover the central 5.5 and 8.3 dva, respectively(Hendrickson, 2005). Since our probe stimulus exhibited a diameter of 3 dva and therefore extended past the foveola and into the surrounding foveal region, we use the term ‘fovea’ to refer to observers’ center of gaze throughout the manuscript. To facilitate the integration of our findings into existing literature(Stewart et al., 2020), visual field locations outside the parafoveal area will be referred to as ‘peripheral’.”

One paper of potential relevance for the discussion is "Transsaccadic integration of visual information is predictive, attention-based, and spatially precise" by Wilmott and Michel.https://jov.arvojournals.org/article.aspx?articleid=2776566

We thank the reviewer for reminding us of this reference. The idea of separating trials depending on the generated response and determining the mean stimulus properties presented on a certain subset of trials is indeed similar to our approach of determining the mean noise properties when a congruent or incongruent FA had been generated. We added this reference to the Results section where we initially describe this approach:

“On each trial, we identified all noise images that had been displayed during the potential probe presentation period, i.e., from the onset of the dynamic noise stream to saccade onset (for a conceptually similar approach using luminance-modulated patches see Wilmott and Michel, 2021).”

We have furthermore added the reference when explaining the concept of predictive remapping to cite a range of methodologies that support the existence of this mechanism (section ‘Enhancement is aligned to the center of gaze, not to the remapped target location‘ in Results).

[Editors' note: further revisions were suggested prior to acceptance, as described below.]

The manuscript has been improved but there are some remaining issues that need to be addressed, as outlined below:All three reviewers appreciated the manner in which you were able to address the great majority of their comments, and the additional experiments performed, which were deemed to add significant value to the study.Reviewers were reasonably convinced by your arguments that criterion shifts are unlikely to explain the phenomena at hand. However, they requested that you modify your Discussion to include consideration of alternative mechanisms. Specifically, please respond to the following concerns, which revolve around the notion that dismissing criterion shifts as explaining the obtained results because of spatial/temporal specificity might be too simplistic an interpretation:1. Please discuss whether the particular task design used in the present study could cause spatial distributions of hit rates and false alarm rates that might be incorrectly interpreted as enhancement, as suggested in the work of Sridharan et al., JNeurosci (2017).

To address this question, we reviewed both the referenced paper and the original studies included in the simulations. Sridharan et al.’s model assumes a location-specific shift in choice criterion and explains a range of behavioral findings in tasks involving spatial components. This also constitutes the decisive difference between these studies and our design: while changes in criterion according to Sridharan et al. (2017) would affect any visual information presented at a certain visual field location, our results would require an interplay between spatially specific and feature specific criterion variations. Only a criterion shift that selectively affects orientation information which (a) appears in or near the center of gaze and (b) matches the orientation of a peripheral stimulus could account for our findings. Even if these variations are accomplished through an as of yet unknown mechanism, the results of our parafoveal control condition, in which we found no difference between congruent and incongruent Hit Rates despite spatial predictability, would remain unaccounted for.

2. Additionally, based on the extensive literature on temporal changes in criterion during decision-making (i.e., "collapsing bounds" or "urgency signals"), please discuss if there could there be urgency signals during saccade preparation that lead up to a decision and then saccade generation? Is it possible that the reason for an increase, then a decrease in hit rates pre-saccadically is a temporal change in criterion until saccade generation is hit (i.e., a bound is crossed)?

We added the interpretation that an increase in Hit Rates between the baseline and earliest presaccadic time point may constitute an urgency signal to the Results section (where we had incorporated a different suggestion during the first revision). We consider it unlikely that the subsequent, continuous decrease in Hit Rates across the saccade preparation period reflects a change in criterion since a similar decrease in foveal performance has been reported in an orientation discrimination task that did not allow for systematic criterion effects. We added a more extensive explanation to the Discussion.